PLOS One logo

# Novel insights from comprehensive analysis: The role of cuproptosis and peripheral immune infiltration in Alzheimer's disease

Jing Wang[1,2☯], Zi-Wen Yu[1,2☯], Qi Liu[1,2], Jing-Xun Wu[3], Yi-Dan Zhang[1,2], Hui-Juan Wan[1,2], Min Bi[1,2], Nai-An Xiao[1,4]*, Kun-Mu Zheng[1,2]*, Bin Jiang[1,2]*

**1** Department of Neurology and Department of Neuroscience, The First Affiliated Hospital of Xiamen University, School of Medicine, Xiamen University, Xiamen, Fujian, China, **2** Xiamen Key Laboratory of Brain Center, Xiamen, Fujian, China, **3** Department of Medical Oncology, The First Affiliated Hospital of Xiamen University, School of Medicine, Xiamen University, Xiamen, Fujian, China, **4** Department of Neurology, The Third Hospital of Xiamen, Xiamen, Fujian, China

☯ These authors contributed equally to this work.

* binjiang@xmu.edu.cn (BJ); kumuzheng@163.com (K-MZ); wsxna@163.com (N-AX)

## Abstract

### Background

Cuproptosis is increasingly recognized as an essential factor in the pathological process of Alzheimer's disease (AD). However, the specific role of cuproptosis-related genes in AD remains poorly understood.

### Methods

Our first step was to obtain gene expression data from the GEO database and identify differentially expressed cuproptosis-associated genes (DECAGs) in AD. GO, KEGG, and GSEA analyses were then conducted on these genes. Subsequently, we attempted to classify AD patients by unsupervised clustering. Then, four machine-learning models were used to screen hub-genes from the DECAGs. We also explored the immune features of these genes and predicted target drugs. Molecular docking analysis was then performed on the predicted drugs and their corresponding hub-gene related proteins. Candidate markers were then validated by single-cell analysis and intracellular communication was investigated in a GEO scRNA-seq dataset. Lastly, we examined the expression levels of the hub-genes in peripheral blood cells using real-time quantitative PCR.

### Results

19 DECAGs were found in AD and the key biological processes and molecular functions associated with AD were further determined. Two subtypes of peripheral blood cells showed significant alternations in AD: Cluster1 and Cluster2. Five hub-genes

**Data availability statement:** All relevant data are within the manuscript and its Supporting Information files.

**Funding:** This work was supported by grants from the Natural Science Foundation of Fujian Province to Bin Jiang (No. 2022J011359) and Jing-Xun Wu (No. 2022J011380), the Medical and Health Guiding Project of Xiamen City to Qi Liu (No. 3502Z20224ZD1013) and Jing-Xun Wu (No. 3502Z20214ZD1011). The funders had no role in study design, data collection and analysis, decision to publish, or preparation of the manuscript.

**Competing interests:** The authors have declared that no competing interests exist.

including *FDX1*, *GLS*, *PDK1*, *MAP2K1*, and *SOD1* were then screened out from the machine-learning study. All of the five hub-genes were significantly correlated with various immunocytes. We discovered compounds targeting hub-gene related proteins and forecasted multiple strong hydrogen bonding interactions between the picked predicted drugs and the target proteins by molecular docking analysis. Subsequently, in the single-cell analysis of AD peripheral blood, all hub-genes except SOD1 were found to be up-regulated in B cells, NK cells, and CD4+T cells, possibly acting on the MIF pathway. Finally, we discovered that the levels of *PDK1* expression in AD patients were remarkably upregulated, while *FDX1* and *GLS* were significantly decreased using qPCR.

## Conclusion

This study examined changes in intercellular communication between immune cells in the peripheral blood and identified five novel feature genes associated with cuproptosis in AD patients. These results facilitated a deeper understanding of the molecular mechanisms of AD and suggested novel therapeutic targets.

## Introduction

Alzheimer's disease (AD) is the predominant type of dementia worldwide, and its main clinical symptoms include severe disability and memory loss [1]. The pathology of AD is characterized by intraneuronal neurogenic fiber tangles and extracellular amyloid plaques [2]. Unfortunately, regardless of the research efforts and high prevalence, there are no curative therapies to prevent or slow the progression of the disease, necessitating an urgent need for new therapeutic targets.

Recently, the involvement of neuroinflammation and cuproptosis in AD pathogenesis has received increasing attention [3–5]. Furthermore, interactions between immune cells in the peripheral blood (including macrophages, monocytes, B cells, and T cells) and the brain in AD may hold the key to treatment [6–8]. Multiple categories of peripheral immunocytes may be involved in AD progression, including innate and adaptive immunity [8,9]. The presence of CD8 T lymphocytes in the cerebrospinal fluid (CSF) of patients with AD has been reported, and they may infiltrate the central nervous system via the choroid plexus and are involved in an adaptive immunity response [10]. In AD models, infiltration of circulating monocytes into the brain reduces the Aβ burden and improves cognitive performance, indicating that it plays a beneficial disease-modifying role in AD pathology [11]. However, no studies have reported on exploring peripheral blood adaptive repositories in AD patients.

Cuproptosis is a newfound type of copper-induced cell death in which the excessive accumulation of lipid-acylated proteins may lead to cell death through proteotoxic stress [12]. Copper homeostasis is known to be heavily dependent on the regulatory role of mitochondria [13–15]. Several recent studies have linked tau proteins in AD to mitochondrial malfunctions [16,17]. Overexpression of aggregated and hyperphosphorylated tau may impair mitochondrial dynamics and axonal transport in various

organelles. Moreover, hyperphosphorylated tau impairs axonal transport in mitochondria, thereby damaging synaptic and neural function and leading to memory impairment in AD.

Rapid advances in bioinformatics and sequencing technology have opened another window for elucidating the pathogenesis of AD [18]. Seurat is an R software package for visualizing and analyzing data from single-cell RNA sequencing (scRNA-seq) [19]. The CellChat deduces inter-cellular communication by analyzing transcriptome data from single-nucleus RNA sequencing or scRNA-seq [20]. This approach enables the visualization, analysis, and quantification of intercellular communication networks. Four machine-learning algorithms have been established for the in-depth study of cuproptosis genes involved in AD: generalized linear models (GLM), eXtreme Gradient Boosting (XGB), support vector machine (SVM), and random forest (RF) [21].

In this study, we performed Seurat and CellChat analyses of single-cell sequencing transcriptomic data from peripheral blood of AD patients and normal controls. We found that intercellular communication in immune cells related to cuproptosis was profoundly altered in the peripheral blood. Through machine-learning models, we also identified 5 cuprotosis-associated genes involved in AD pathogenesis and further validated their alterations in our cohort.

## Methods

### Data acquisition and processing

In this study, we harvested the bulk mRNA datasets GSE63060, GSE33000, and GSE122063, along with the scRNA-seq dataset GSE181279 for AD and healthy controls from the Gene Expression Omnibus (GEO) database [22–24]. Detailed information about the data set is provided in S1 Table. The GSE63060 dataset contained 329 blood samples from 145 patients with AD, 80 mild cognitive impairment (MCI) subjects, and 104 matching control subjects. The GSE181279 dataset consisted of immune cell signatures of 36849 peripheral blood cells in AD patients and normal controls. Furthermore, we obtained 46 cuproptosis-associated genes (CAGs) from the GeneCards database (https://www.genecards.org/) and previously published literature, [25] (S2 Table). All datasets were processed utilizing R Software (v4.4.1). When multiple probes in the bulk dataset pointed to one gene, we took its average value as the expression value. Subsequently, we eliminated unwarranted variations and batch effects by the "sva" package, following which the "affy" package was applied to normalize and process these raw gene expression files. Eventually, the "Seurat" package was used to process scRNA-seq data [26].

### Differentially expressed CAGs (DECAGs) analysis

The "limma" package was used to discover differentially expressed genes for AD patients and healthy controls within GSE63060 using the standard $|\log_2$ fold change (FC)$| > 1$ and Bonferroni-p value $< 0.05$. DECAGs were obtained by intersecting with the DEGs and 46 cuproptosis-associated genes. Subsequently, the "pheatmap" and "ggplot2" packages were utilized to represent the results visually.

### Functional enrichment and Protein-protein interaction (PPI) network

We employed the "ClusterProfiler" software package to perform Kyoto Encyclopedia of Genes and Genomes (KEGG) and Gene Ontology (GO) analyses to gain a better comprehension of the biological roles and pathways involved in the DECAGs. Meanwhile, we visualized these results using the Metascape database [27]. The PPI network was constructed using the Linear STRING repository (https://string-db.org/) to explore the interactions between the proteins corresponding to these genes, with 0.900 (maximum confidence level) assigned as the threshold score for interactions.

### Immune characteristics

CIBERSORT is a utility to assess the relative proportions of various cell subpopulations in organisms of large RNA sequencing datasets [28]. More and more studies inferred that many types of immune cells in the circulating blood might

 

participate as part of the occurrence and development of patients with AD. To further investigate this phenomenon, we uploaded the bulk RNA dataset GSE63060 to CIBERSORT to obtain the ratio of 22 infiltrating immune cells in each sample. Of all the cells is equal to 1, and the p-values for every block of convolution were then established empirically. Those samples for which the p-value was less than 0.05 were considered significant. Thereafter, single-sample gene set enrichment analysis (ssGSEA) was performed using the "GSVA" R package to estimate the constitution of immune-related functions in the richness of individual samples [29]. Ultimately, visualization of the proportion of immunocytes in the healthy control and AD samples was performed by a heatmap and box plot.

## Relevance analysis

The correlations and correlation coefficients obtained within DECAG were further investigated utilizing the "corrplot" and "circlize" software packages. Spearman correlation analysis of DECAGs and immunocyte infiltration was then performed by the "ggcorrplot" R package. Based on the correlation coefficients, the effects of interactions with p-values less than 0.05 were considered significant.

## Unsupervised clustering

In order to obtain a thorough overview of the effects of cuproptosis in the progression of AD, we attempted to subtype AD patients with the expression profiles of DECAGs. Cluster analysis of 310 patients with AD was performed using the "ConsensusClusterPlus" software package. We used K-means for the clustering algorithm. The maximum subtype number of K was six. Based on different subsamples, each clustering analysis was repeated 50 times. The percentage of samples used for each resampling was 0.8. We used all features for each resampling (pFeature = 1). The distance measure was "euclidean". The seed was set as "123456".

The consensus matrices and the cumulative distribution function (CDF) curve were used to determine the optimum number of clusters. Furthermore, the optimum clustering results were intuitively visualized using principal component analysis (PCA) to verify their reasonableness and stability.

## Gene set variation analysis (GSVA)

GSVA has been extensively used for exploring the biological functions and pathways of enriched gene sets in constructed clusters. Therefore, we conducted GSVA of the DECAGs-related clusters utilizing the "GSVA" and "GSEABase" packages. The referring sets "c5.go.symbols", "c2.cp.kegg.symbols", and "c7.immunesigdb.v2023.1.Hs.symbols" were downloaded out of the MSigDB database. Significant differences were found for absolute values of t greater than 2.

## Construction of cuproptosis-related AD risk models

Firstly, the 145 AD patients in GSE63060 dataset were randomly divided into a test cohort (30%, n = 44) and a training cohort (70%, n = 101). Next, we trained the expression profiling the genes of the DECAGs by employing the Xtreme Gradient Boosting (XGB), generalized linear models (GLM), random forest (RF), and support vector machine (SVM) to forecast the AD patients' clinical outcomes. The XGB automatically selects the best number of features from all collected attributes based on the weights to select the optimal number of features to predict a particular outcome [30]. The GLM is an improved version of the multivariate logic model that offers a flexible assessment of the link between a continuously independent feature and an ordinally distributive dependent variable feature [31]. The RF algorithms are an integrative machine-learning model that utilizes multiple independent decision treaties to identify the optimal number of variables [32]. The SVM locates the optimum variable by finding the match with the minimum cross-validation error [33]. We used the "caret" (version 6.0.94), "randomForest" (version. 4.7.1.2), "kernlab" (version.0.9.33), "xgboost" (version.1.7.8.1), and "e1071" (version.1.7.16) packages to perform the machine-learning models described above. Optimal selection was

ensured using 5-fold cross-validation and performed with default parameters. The diagnostic performance of the above approaches was then assessed through residual analysis using the "model_performance" function, ROC curves using the "pROC" package, and feature importance analysis using the "variable_importance" function. An optimal technique was consequently established for which the first five important genes were adopted as critical diagnostic indicators of AD risk, defined as hub-genes. A nomogram was created by performing a multifactorial logistic regression on these 5 genes utilizing the "rms" (version 6.8.2) R package. Every gene has a matched "Points" together with a "Total Points" representing a sum of the points of the reference gene. The forecasting capability was estimated with calibration, ROC curves, and decision curve analysis (DCA). Lastly, we employed GSE33000 and GSE122063 as external datasets and validated the risk model using ROC curves. Among them, GSE33000 contains 310 AD and 157 control brain tissues, while GSE122063 contains 56 AD and 44 control brain tissues.

## Immune analysis, targeted drugs and genes

Subsequently, we explored the associations of hub-genes and immune-infiltrating cells using the "CIBERSORT" and "linkET" algorithms. The Drug-Gene Interaction Database (DGIdb, https://dgidb.org), a comprehensive database for gene-related targeted drugs, served as a drug signature resource to identify potential gene-drug interactions [34]. We uploaded the five hub-genes into the 'Gene/Drug' platform. Gene-drug pairs with interaction scores > 1 were considered significant. GeneMANIA is a powerful online tool and database focused on gene function prediction and network analysis of inter-gene relationships. It helps researchers infer gene function based on existing biological datasets, explore gene association networks, and support the generation and validation of biological hypotheses. We explored the potential targeted genes of the five hub-genes using GeneMANIA database [35].

## Molecular docking

Molecular docking experiments were performed to further validate the intermolecular interactions between the hub-genes and the previously predicted drugs by scrutinizing their binding modes and affinities. Firstly, we picked the most highly scored predicted drugs for interaction in each target protein relating to the hub-gene [36,37] and searched the relevant literature for support [38–40]. Then, we obtained their three-dimensional structures from the PubChem database (https://pubchem.ncbi.nlm.nih.gov/). Three-dimensional structures of the target proteins were then downloaded from the RCSB Protein Data Bank (PDB) database (https://www.rcsb.org/). The crystal structures with high resolution were preferred as the molecule-to-molecule receptors. Subsequently, PyMOL 2.6.0 was utilized to remove water and irrelevant ligands. Finally, binding affinity and pose were predicted using PyRx software, while the results were visualized and analyzed using Discovery Studio to determine protein-ligand interactions.

## Single-cell analysis

We processed scRNA-seq data (GSE181279) analysis with the package "Seurat" (version.5.0.1) [19]. Firstly, this dataset was subjected to quality control, specifically screening for cells that met the following standard: 1) genes with less than three cells; 2) cells with a gene count less than 100 or more than 6000; 3) cells with an RNA count higher than 30000. 4) mitochondrial ratio exceeding 20%. Following this, the scRNA-seq data was normalized utilizing the "NormoralizeData" function to make gene expression in different cells comparable. The top 1500 highly variable genes were then identified using the "FindVariableFeatures" function. Further, we standardized the data by removing the effects of the cell cycle using the "ScaleData" function and performed PCA analysis. The "harmony" function was used to make batch effect corrections. K-nearest neighbor (KNN) graph was constructed using "FindNeighbors" function at 2 and 20 dimensions. After that, we performed clustering analysis utilizing "FindClusters" function with 0.6 resolution. The results were then visualized by conducting t-distributed stochastic neighbor embedding (t-SNE) profiling. The "FindAllMarkers" function was then utilized to identify DEGs, of which the top 10 are shown in the additional heatmap and used for annotation clustering

via the "SingleR" function. After that, we grouped the identified major cell types according to the type to which the samples belonged (AD or control) and visualized the results using t-SNE plots. Finally, we examined the previously mentioned 5 hub-genes for their levels of expression in the identified cell types and represented them in density plot, scatter plot, and bubble plot.

### Cell-cell communication and signaling pathway analysis

CellChat(version 1.6.1), as an algorithm for analyzing interactions among ligands, receptors, and cofactors that incorporates a ligand-receptor database, was employed to characterize the key signals as well as the intercellular communication networks from the annotated cell clusters [41]. Next, the "computeCommunProb" function, as well as the "computeCommunProbPathway" function in CellChat, were conducted to calculate potential cell-cell communication and signaling pathways with default parameters, meanwhile, the interactions among less than 10 cells were filtered out utilizing the "filterCommunication" function. Furthermore, the "aggregateNet" function was conducted to quantify the amount of ligand-receptor as well as intercellular communication strength among clusters of cells. At last, the "NetAnalysis _" and the "NetVisual _" functions were employed to depict the interested signaling pathways.

### Clinical sample collection and evaluation

The First Affiliated Hospital of Xiamen University Medical Research Ethics Committee reviewed and approved the research involving human participants (**No.EC-2024-XMU-1619**). All participants gave informed written consent to participate in this study. From March 1, 2024 to October 1, 2024, we recruited 6 patients aged 60–82 years who were diagnosed with probable AD according to National Institute of Aging-Alzheimer's Association (NIA-AA) criteria and a positive amyloid (18F-florbetapir) positron emission tomography (PET) scan from our Department of Neurology and 4 matched healthy controls with negative amyloid PET from our Physical Examination Center. All participants gave informed written consent to participate in this study. Other inclusion criteria for AD patients were: (1) Mini-mental state examination (MMSE) ≤26 points; (2) Modified Rosen-Hachinski Ischemic Scale (HIS) ≤4 points. As for healthy controls were: (1) MMSE ≥27 points; (2) HIS ≤ 4 points. Participants with a family history of dementia, with other medical conditions or drug usage that may affect cognition, were excluded [42]. The demographic characteristics of these individuals are described in Table 1. All participants or their primary caregivers provided informed permission to be involved in this trial, and the ethical commission of our hospital assessed their eligibility to participate following the Declaration of Helsinki. We collected peripheral blood from all participants using PAXgene blood RNA tubes (Qiagen, Hilden, Germany) and strictly followed the instructions of the manufacturer for storage.

### Real-time quantitative PCR (RT-qPCR)

Isolation of total RNA from the samples was performed utilizing TRIzol Reagent (Service bio, Hangzhou, China) according to the producer's protocol. We then performed reverse transcription and qPCR analysis was carried out with SYBR Green qPCR Master Mix (AG11746). *GAPDH* served to be the inner reference gene, and the sequence of primers was showed in Table 2. All laboratories were performed in triple copies. Quantification of *FDX1*, *GLS*, *MAP2K1*, *PDK1*, and *SOD1* relative levels of expression was done with the two^-ΔΔCT method. Data were statistically analyzed using GraphPad Prism (version 9.5), and Student's t-test was used to estimate differences between groups. The definition of statistical significance is a p-value of less than 0.05.

### Statistics

Bioinformatics analyses were performed using the mentioned packages in the R (version 4.3.1), and we analyzed the data from the in vitro experiments using the GraphPad Prism 9.5 software. The correlation was determined by Spearman correlation analysis. For comparison, the Student's t-test worked for normal distribution variants, whereas the Wilcoxon

**Table 1. Summary of patient demographic characteristics.**

| Code | Diagnosis | Sex | Age | Education (y) | MMSE | MoCa | HIS |
|---|---|---|---|---|---|---|---|
| AD1 | AD | male | 67 | 12 | 19 | 15 | 1 |
| AD2 | AD | male | 74 | 16 | 22 | 16 | 2 |
| AD3 | AD | female | 69 | 12 | 15 | 13 | 2 |
| AD4 | AD | male | 61 | 9 | 17 | 15 | 2 |
| AD5 | AD | female | 77 | 9 | 20 | 18 | 1 |
| AD6 | AD | female | 82 | 16 | 18 | 15 | 1 |
| HC1 | Healthy | female | 60 | 16 | 30 | 28 | 0 |
| HC2 | Healthy | female | 67 | 16 | 30 | 29 | 0 |
| HC3 | Healthy | male | 69 | 12 | 30 | 28 | 0 |
| HC4 | Healthy | male | 66 | 12 | 30 | 29 | 0 |

Abbreviations: AD, Alzheimer's disease; HC, healthy controls; F, female; M, male; MMSE, mini-mental State Examination; MoCA, Montreal Cognitive Assessment. HIS, modified Rosen-Hachinski Ischemic Scale.

**Table 2. Primers of the tested genes.**

| Genes | Primers (forward) | Primers (reverse) |
|---|---|---|
| SOD1 | 5'-GGTGTGGCCGATGTGTCTATT-3' | 5'-TCC-AGCGTTTCCTGTCTTTGTA-3' |
| FDX1 | 5'CCGGAGCAGCTCAGAAGATAA-3' | 5'-TTTCAGGCACTCGAACAGTCAT-3' |
| PDK1 | 5'-TGGTTT-TGGTTATGGATTGCC-3' | 5'-GCCTCGTGGTTGGTGTTGTA-3' |
| GLS | 5'-ATGATGTGCTGGTCTCCTCCT-3' | 5'-GTTTGATTTTCCTTCCCGTTG-3' |
| MAP2K1 | 5'-GTGGGCTTCTATGGTGCGTT-3' | 5'-AATGAGTCCCCTGGAGTCTTTC-3' |
| GAPDH | 5'-CTCTCTGCTCCTCCTG-TTCGAC-3' | 5'-GCCCAATACGACCAAATCCG-3' |

rank-sum test was applicable to non-normal distribution variants. For statistical significance, a Two-tailed p-value of < 0.05 was used.

## Results

### Analysis of AD-related DECAGs

The basic framework is shown in Fig 1. By intersecting with 46 cuproptosis-associated genes, 19 DECAGs were filtered from the GSE63060 dataset totally (Fig 2A). Of these, *COX17, ATP7A, MTF1, ULK2, PDK1, SOD1, MT1A,* and *MT2A* were upregulated in AD groups, while *FDX1, GLS, MAP2K1, SCO2, SCO1, PDHB, UBE2D4, PDHA1, UBE2D2, DLD,* and *DLAT* were downregulated (Fig 2B). The network further demonstrated the close connections of these genes (Fig 2C). The correlation coefficients between these DECAGs are shown in Fig 2D. Afterwards, enrichment analyses of these DECAGs were conducted on the Metascape database (Fig 3A and 3B). DECAGs were significantly enriched in biological process modules related to acetyl-CoA biosynthetic and metabolic processes as GO analysis disclosed. For cellular component and molecular function modules, these genes showed enrichment in alpha-ketoacid dehydrogenase complex, oxidoreductase complex, oxidoreductase activity, and copper ion binding (Fig 3C, S3 Table). The analysis of the KEGG pathway suggested that most of these genes were involved in the HIF−1 signaling pathway and Lipoic acid metabolism, among others (Fig 3D, S4 Table). Furthermore, protein interactions between DECAGs were made available for visualization by Cytoscape to present sub-networks that exhibited high correlations (Fig 3E). Enrichment analyses for disease also showed that these genes were strongly correlated with Pyruvate Dehydrogenase Complex Deficiency Disease, Cryptogenic Chronic Hepatitis, and Hypocupremia (Fig 3F).

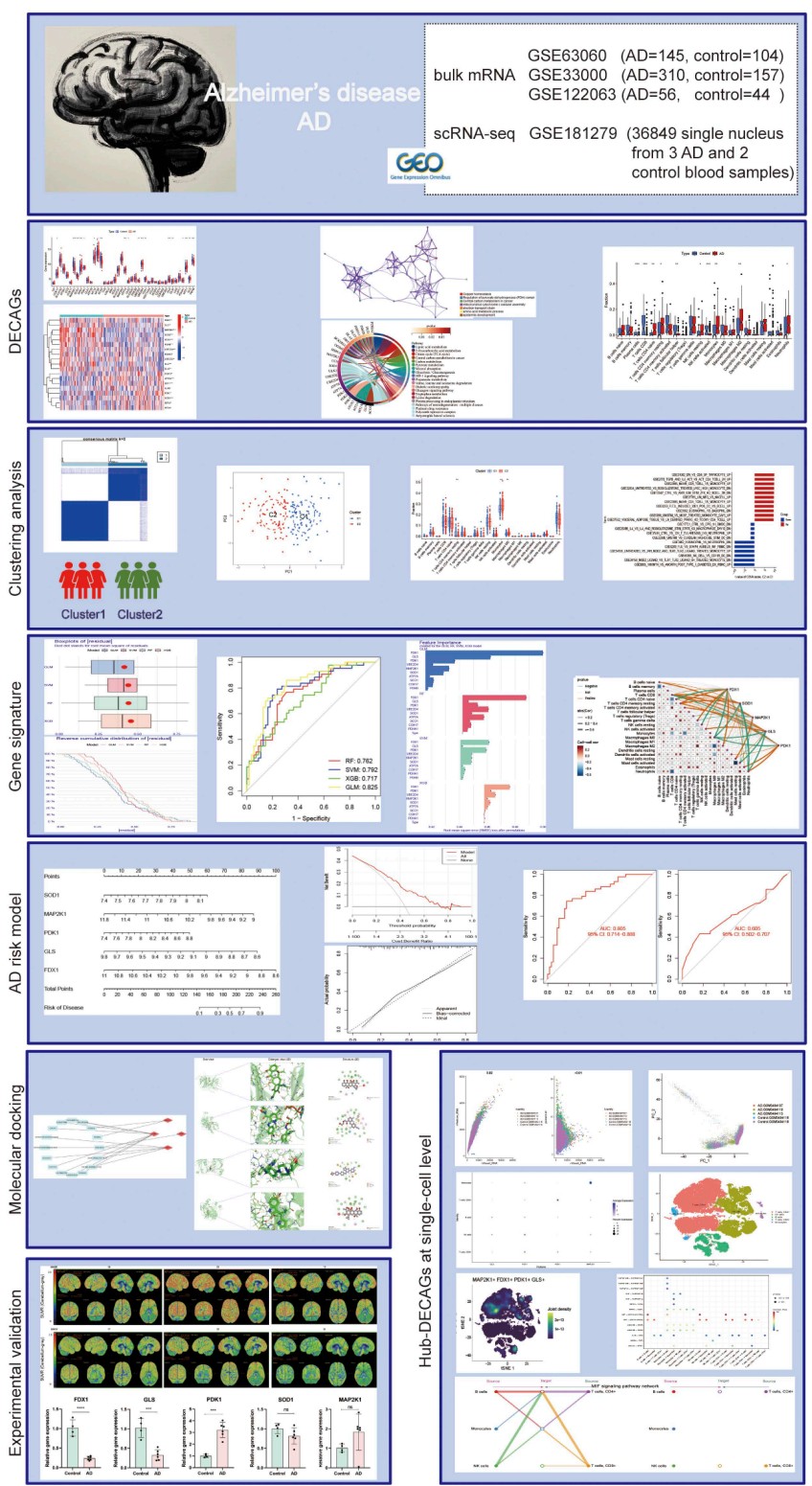

**Fig 1. The basic flowchart of analysis in this study.**

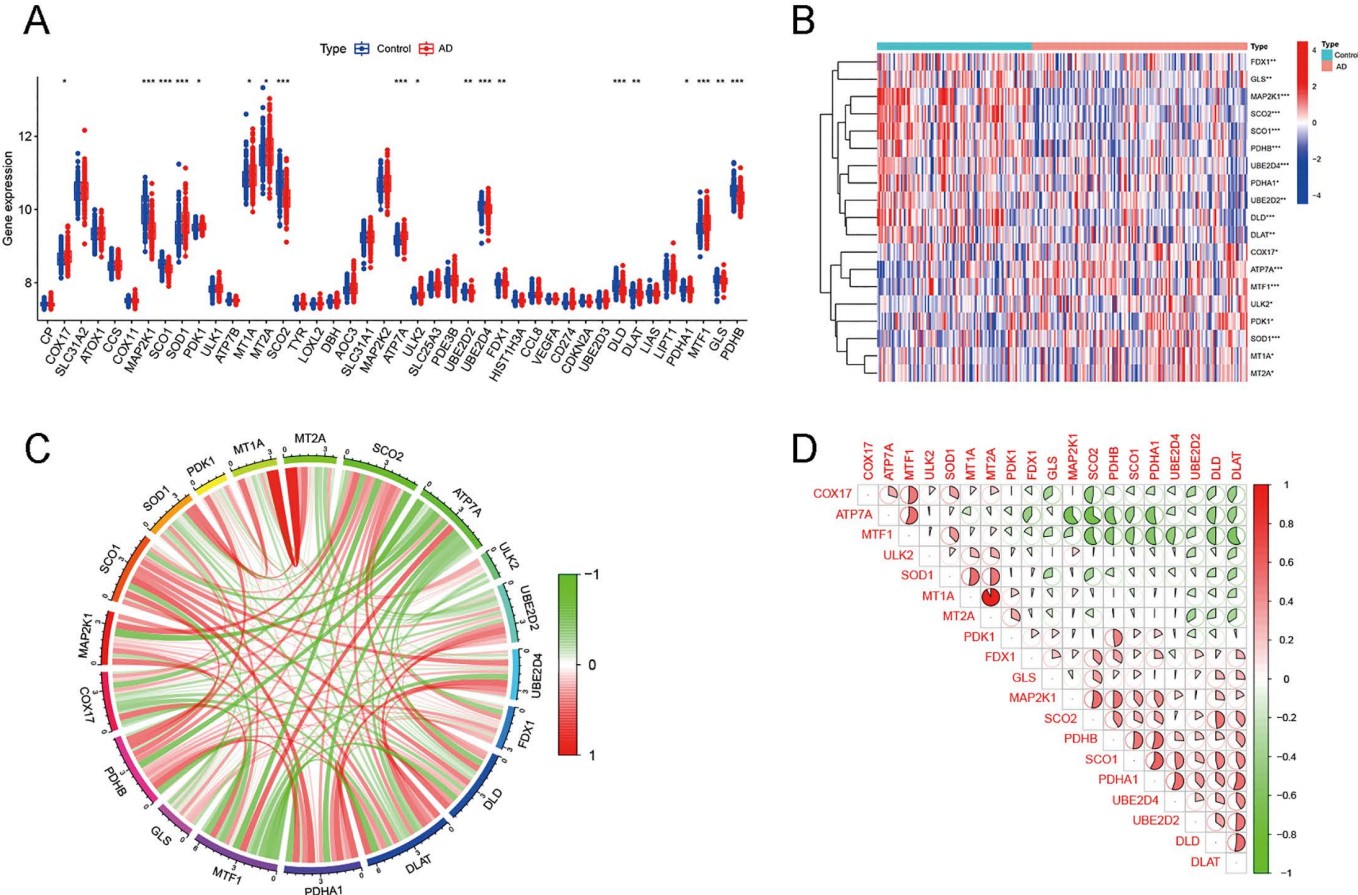

**Fig 2. The expression profiles of 19 DECAGs between AD patients and normal controls. (A)** The boxplot showed the expression characteristics of 46 CAGs in AD. **(B)** The heatmap of DECAGs between AD and normal samples. **(C)** Ring diagram of the gene relationship network of 19 DECAGs, with red and green representing positive and negative correlations, respectively. **(D)** Correlation coefficient analysis of 16DECRG, expressed as the area of pie charts.

## Relationship between DECAGs and immune cells

Comparison and analysis of infiltrating immune cells in AD samples and healthy controls is often performed by CIBER-SORT [43]. Fig 4A shows the differential composition of immune cell infiltrates in the two groups. The AD group exhibited greater ratios of NK cells, B cells, T follicular helper cells, eosinophils, CD8 T cells, and M0 macrophages compared with the controls. In contrast, the ratios of infiltrating Treg, neutrophils, activated mast cells, and T gamma delta cells were reduced in comparison with healthy controls (Fig 4B). Moreover, Pearson analysis was conducted on various immune cell types (Fig 4C). The results showed that CD8 T cells and CD4 memory resting T cells exhibited a relatively strong negative correlation (correlation coefficient = −0.42). And M0 macrophages and activated mast cells showed a greater positive correlation (correlation coefficient = 0.29).

## Clustering analysis

Clustering of DECAGs based on expression differences between groups allows genes with greater expression homogeneity within clusters to be gathered into a single cluster. The results showed that 3 or more clusters were not powerful

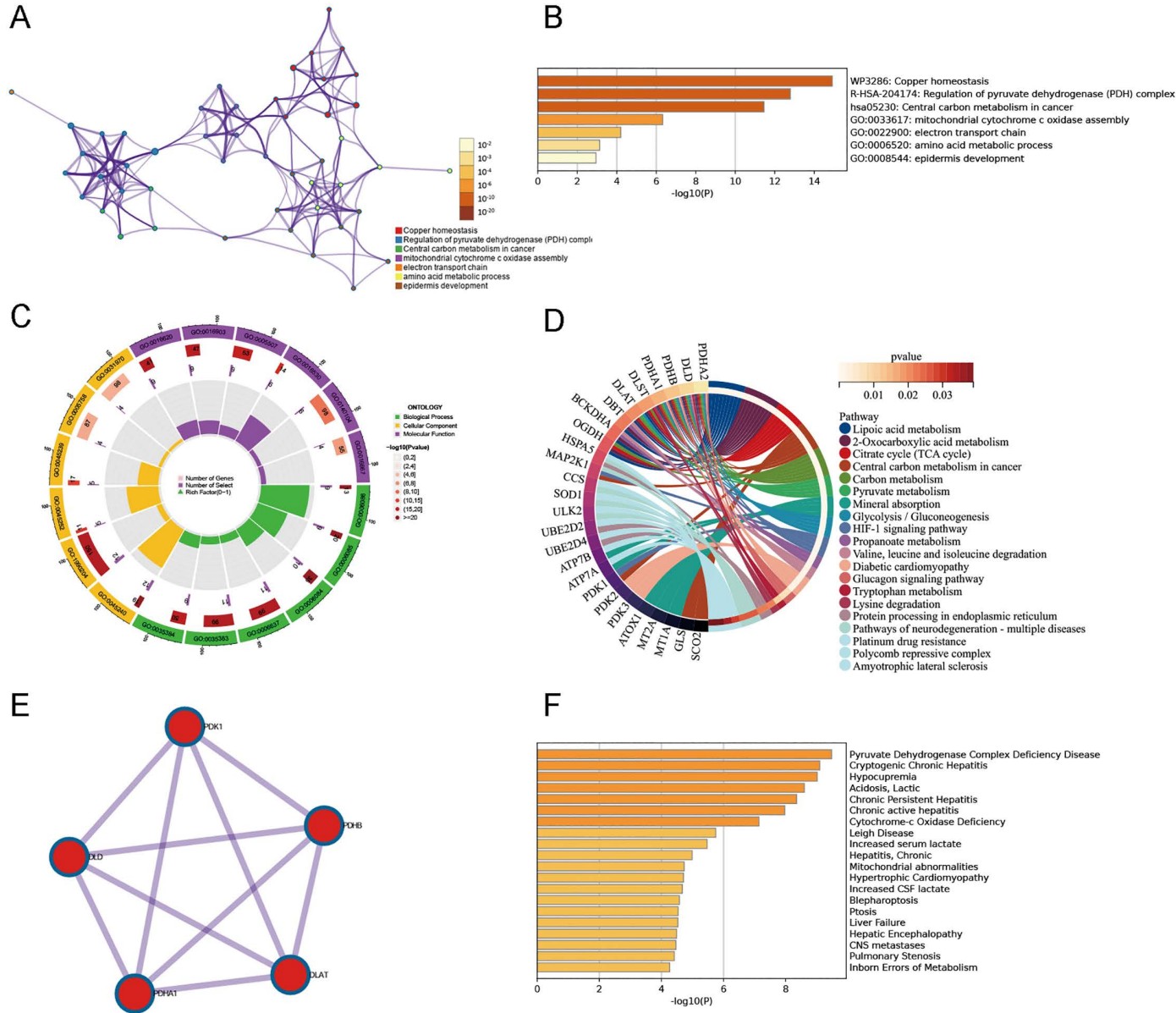

**Fig 3. Functions and pathways enrichment analysis of the 19 DECAGs in AD. (A)** Networks of GO and KEGG enriched terms using the Metascape database. **(B)** P-value-based bar charts of biological processes. **(C)** The circle plot of GO analysis results. **(D)** Circle diagram of the KEGG pathway results. **(E)** The constructed PPI network using the 19 DECAGs with interaction score set at >0.900. **(F)** The significant enriched disease terms of the 19 DECAGs.

enough to distinguish subgroups, so we chose to cluster the AD samples into 2 subgroups (Fig 5A–5C, S5 Table). To test the clustering effect, we found that the 2 subgroups could be well differentiated using principal component analysis (Fig 5D). The blue points represented Cluster1 (n = 90), and the red points represented Cluster2 (n = 55). To further understand the expression of these DECAGs between the clusters, we performed differential analysis, which revealed that COX17, SOD1, PDK1, MT1A, MT2A, ULK2, and MTF1 were upregulated in Cluster2, while GLS, DLD, DLAT, UBE2D2 and SCO2 were downregulated in Cluster2 (Fig 5E). The expression of immune cells between both clusters was then analyzed. The

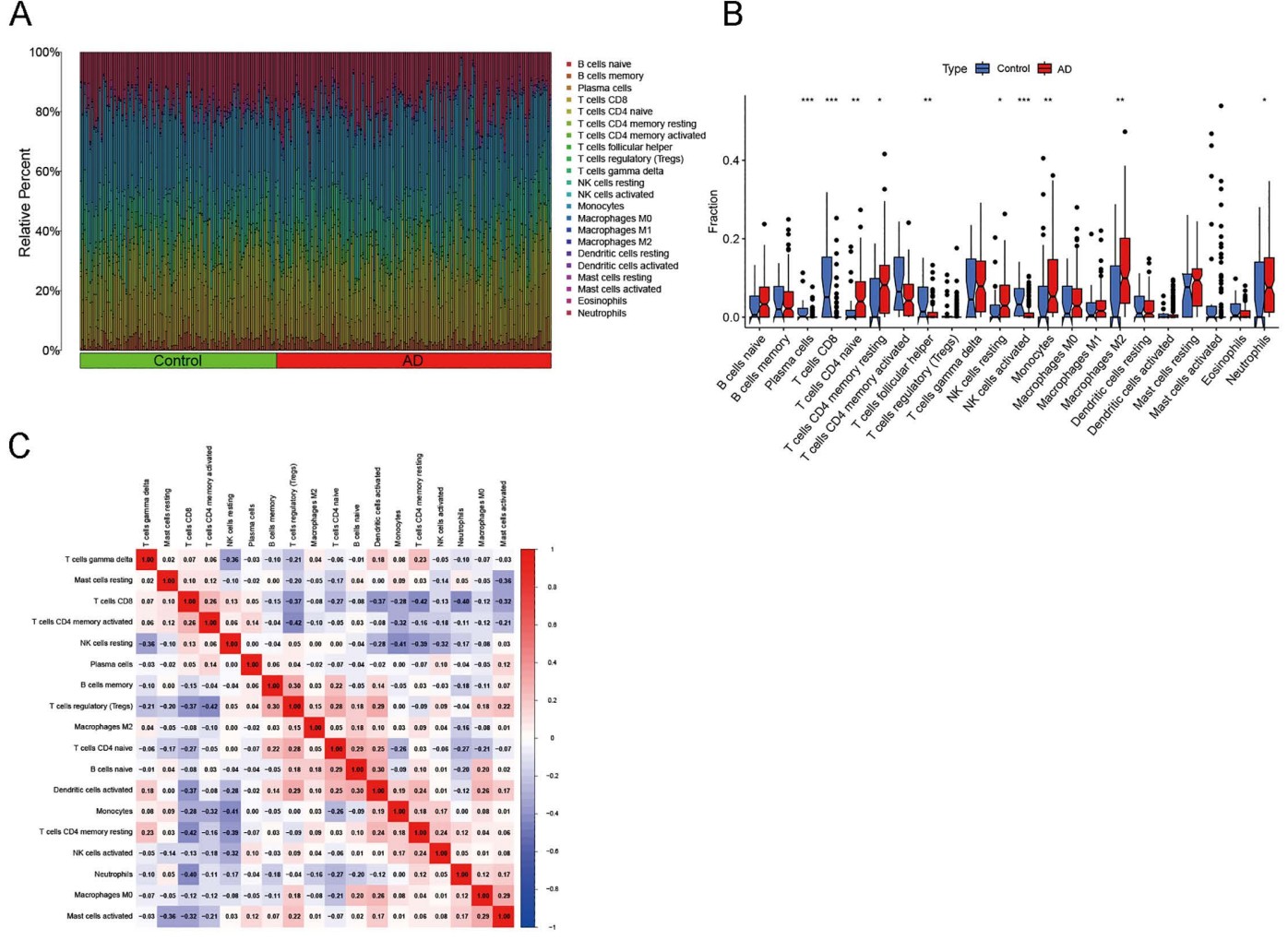

**Fig 4. The immune-infiltrating features associated with cuproptosis in AD. (A)** Relative abundance of 22 immune-infiltrating cells in AD and normal samples. **(B)** The boxplot showing variations in immune microenvironment between AD and normal samples. **(C)** Correlation analysis between the 22 immune-infiltrating cells.

results revealed little variation in immunologic infiltration for the 2 clusters. Monocyte and M0 macrophage levels showed increased infiltration in Cluster2 while resting memory CD4 T cells were less infiltrated in Cluster2 (Fig 5F).

## GSVA

Gene set variation analysis revealed the activated biological pathways across the two clusters. In Cluster1, long-chain fatty acid transport, microtubule nucleation, and enteric nervous system development were reinforced. At the same time, Cluster2 was identified to be closely associated with protein transmembrane transport and mitochondrial translation (Fig 6A). KEGG analysis revealed that glutathione metabolism, retinol metabolism, and WNT signaling pathway were more active in Cluster1, whereas protein export, spliceosome, and nucleotide excision repair were upregulated in Cluster2 (Fig 6B). Furthermore, we used the reference set "c7.immunesigdb.v2023.1.Hs.symbols.gmt" to investigate the variations in immunorelevant processing for the two clusters (Fig 6C). CD4 SP thymocytes and naive CD8 T cells were more activated in Cluster2. In Cluster1, IL-4 and rosiglitazone macrophages, eosinophils, and neutrophils were more active.

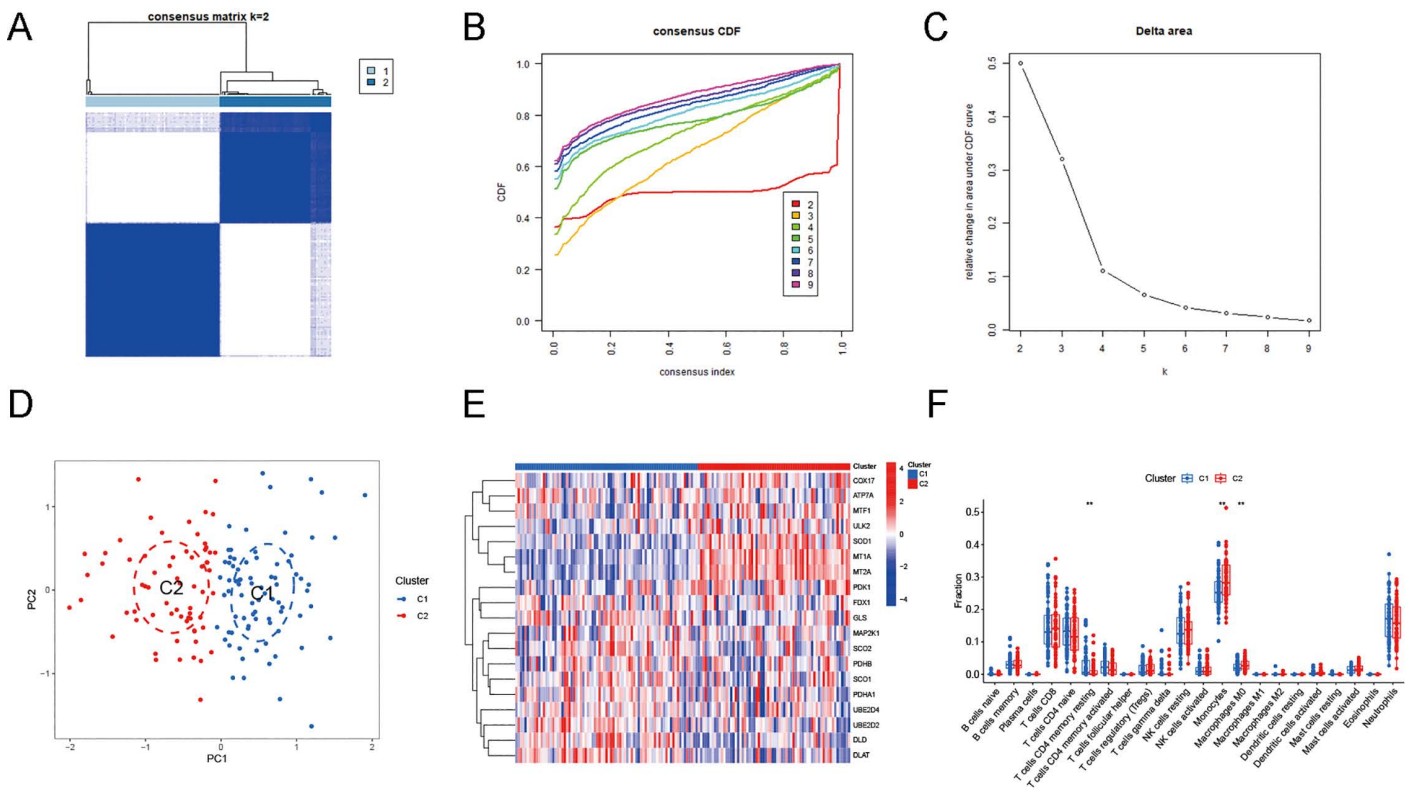

**Fig 5. Consensus clustering analysis based on the DECAGs. (A)** The clustering matrix is most stable when k = 2. **(B)** Cumulative distribution function (CDF) curves when k = 2-9. **(C)** Delta area curves when k = 2-9. **(D)** Principal component analysis of the samples in two clusters. **(E)** The expression heatmap of 19 DECAGs between the two clusters. **(F)** The boxplot showing differences in immune-infiltrating cells between the two clusters.

## Construction and validation of a risk model

Four machine learning algorithms were used to identify the hub DECAGs most closely related to AD. We randomly divided the AD samples, 30% for a test cohort and 70% for a training cohort. GLM and SVM models displayed reduced root mean square of residuals in the training cohort (Fig 7A and 7B). ROC curves analysis of the test cohort validated that the GLM model had the maximum area under the curve (AUC = 0.825, Fig 7C). Thus, we chose the GLM model as the best algorithm for screening AD risk hub-genes. Subsequently, we sorted the most remarkable genes in each model according to the root mean square error (Fig 7D). The first 5 genes ranked in the GLM model: *FDX1*, *GLS*, *MAP2K1*, *PDK1*, and *SOD1,* were selected as variables predicting AD. A nomogram model was built on the basis of these genes (Fig 7E). Each of the five genes was given a different point value, and the points were summed to give the total points. If the total points did not exceed 140, the AD risk was less than 0.1, and if the total points exceeded 240, the AD risk was greater than 0.9. The decision curve analysis (DCA) showed promising clinical benefits related to this nomogram in the training dataset (Fig 7F). Calibration curves revealed high concordance between actual measurements and predicted values (Fig 7G). In addition, we validated the risk model using ROC curves on two external human cortical datasets. The AUC values of the two datasets reached 0.805 (GSE33000) and 0.605 (GSE122063) with 95% confidence intervals (CI) of 0.714–0.888 and 0.502–0.707, respectively (Fig 7H and 7I). These results revealed that our risk model performs well in both blood samples and brain tissue, suggesting that the model was generalizable, robust, and reproducible.

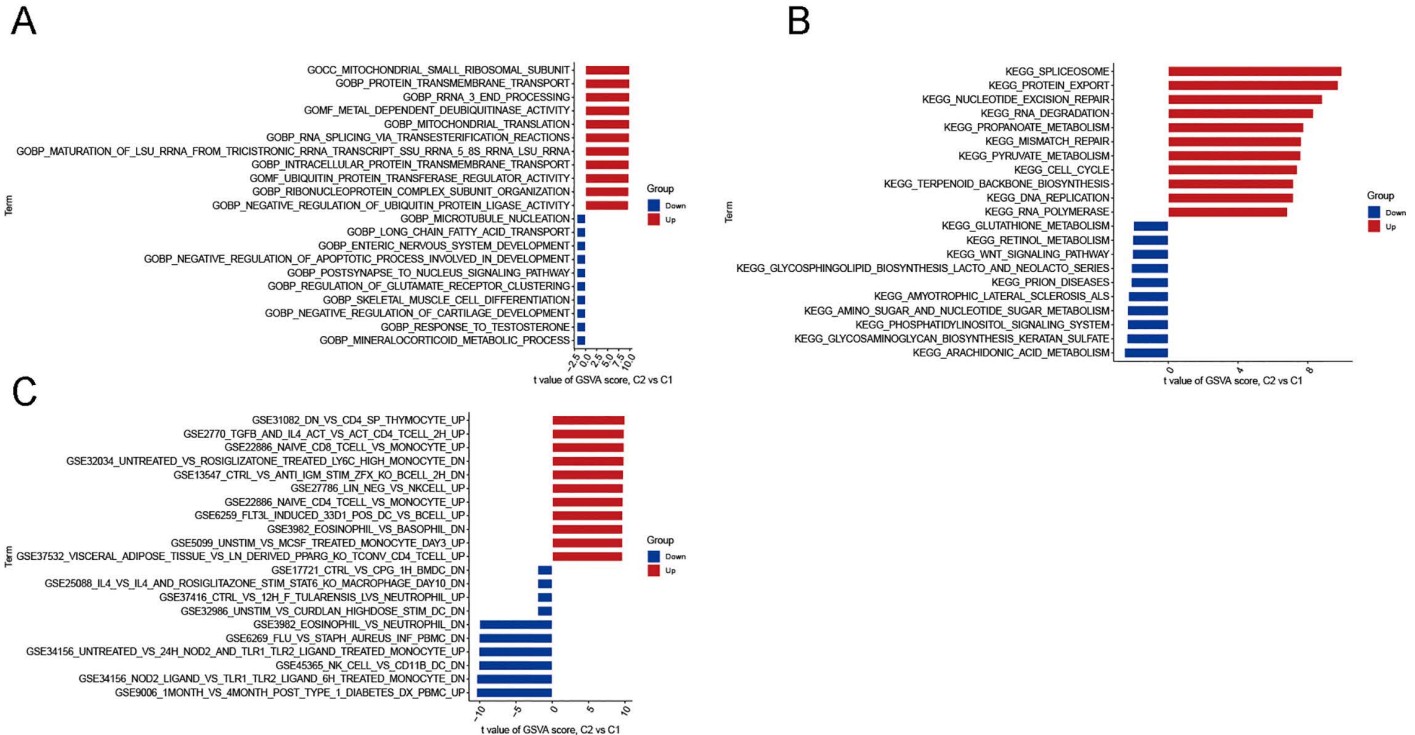

**Fig 6. Gene set variation analysis between the two clusters. (A)** Variations of GO analysis between Cluster1 and Cluster2. **(B)** Differences in KEGG pathways of the two clusters. **(C)** Differences in immune-related pathways between the two clusters.

## Hub-gene associated pathways

An analysis of single-gene GSEA enrichment was performed to exploit the activation pathways between each hub gene's highly and lowly expressed groups. It was found that the high *FDX1* expression was strongly correlated with cytoplasmic translation, whereas its low expression was closely related to the actin cytoskeleton (Fig 8A). Similar to *FDX1*, high *MAP2K1* expression was strongly associated with ncRNA processing. The low expression of *MAP2K1* was correlated with external encapsulating structure (Fig 8B). The high expression of *PDK1* was associated with gas transport, while its low expression was associated with positive regulation of immune response (Fig 8C). Furthermore, the high *GLS* expression might activate the cytoplasmic translation, whereas its low expression was closely associated with a response to wounding (Fig 8D). The high *SOD1* expression was intimately linked to DNA replication and oxidative phosphorylation, while its low expression was tightly associated with complement and coagulation cascades (Fig 8E). We then investigated the biological functions of each hub gene between the highly and lowly expressed groups. The results revealed significant degrees of similarity between the biological pathways of the gene ontology and KEGG pathway analysis (S1 Fig).

## Hub-gene related immune characteristics, drugs, and genes

As shown in Fig 9A, we investigated the relationships between hub-genes and peripheral immunocytes. The results demonstrated that all of the five hub-genes were significantly correlated with various immunocytes, especially with CD8 T cells, CD4 T cells, macrophages, mast cells, and neutrophils, which revealed that all of the hub-genes were involved in regulating peripheral immunity in AD. Then, we predicted drugs that might potentially target these 5 hub-genes with the online DGIdb database (S6 Table) and demonstrated the outcomes using Cytoscape (version 3.8.0). We discovered 2 potential compounds targeting *MAP2K1*, 5 targeting *PDK1*, 8 targeting *FDX1*, and 1 targeting *SOD1* (Fig 9B). To

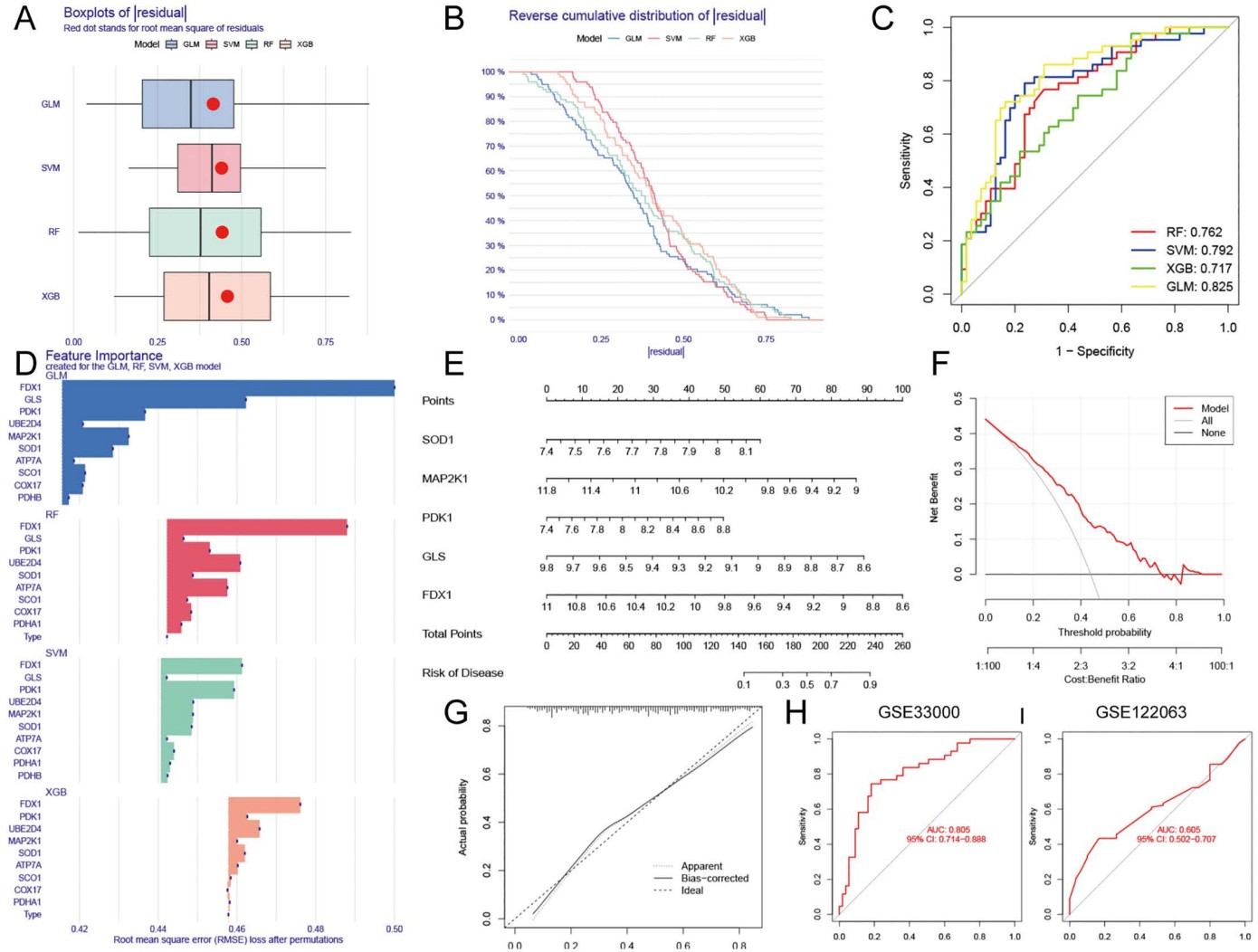

**Fig 7. Establishment of the cuproptosis risk model by integrating multiple analyses. (A)** Area under the ROC curves of each machine learning model. **(B)** The boxplot of residuals in each machine learning model. **(C)** Reverse cumulative distribution of residuals for each machine learning model. **(D)** The significant feature genes created for the GLM, RF, SVM, and XGB models. **(E)** A nomogram created for the five hub genes in GLM model. **(F)** Decision curve analysis displaying the predictive ability of the nomogram. **(G)** Calibration curves showing biases between ideal probability and actual probability. The AUC values were 0.805 and 0.605 for GSE33000 and GSE122063, respectively.

investigate other genes that might potentially influence the 5 hub-genes, we acquired 20 promising genes from GeneMA-NIA that interplayed with the hub-genes (Fig 9C).

## Molecular docking of predicted drugs toward hub-gene related proteins

Molecular docking is primarily a conceptual modeling tool for exploring intermolecular interactions and predicting their binding affinities and modes [44]. To further investigate the binding interactions of the predicted drugs above, we selected the compounds with the highest interaction scores in each hub-gene for protein-ligand molecular docking analyses (Table 3). The binding energies of all 4 picked compounds with the corresponding hub-gene related proteins were below −7.0 kcal/mol, which indicated strong binding affinity between the proteins and the picked predicted drugs [45–49]. According

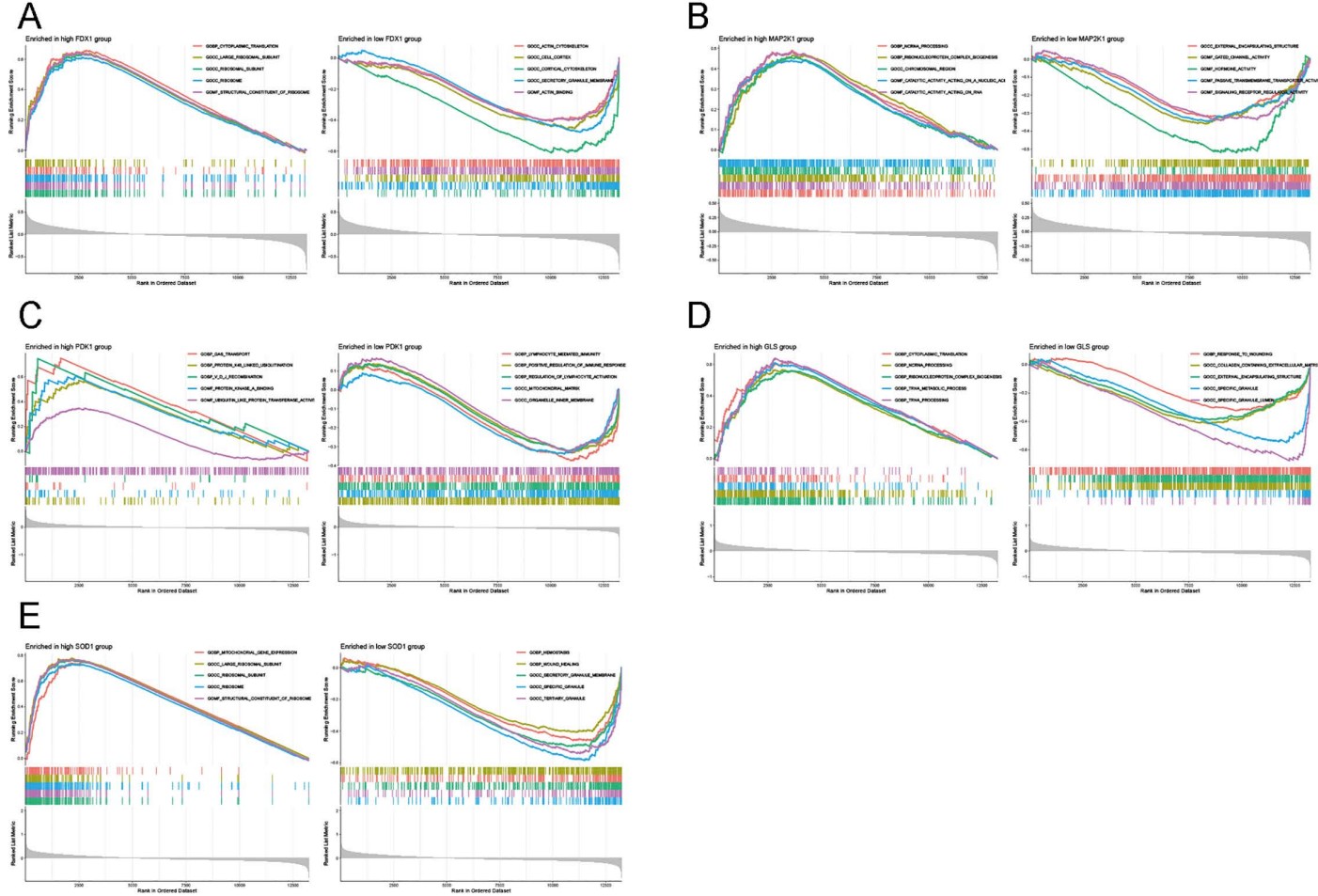

**Fig 8. Gene set enrichment analysis for the five genes in AD patients.** Gene ontology analysis revealed biological functions in the high and low expressions of FDX1 **(A)**, MAP2K1 **(B)**, PDK1 **(C)**, GLS **(D)**, and SOD1 **(E)**.

to the intermolecular interaction visualizations of these molecules, we found that Tetracycline established conventional hydrogen bonds with the amino acids SER-33、ILE-6 in SOD1, the same observation was noted between PDK1 and Dichloroacetic Acid. AZD8330 formed stable hydrogen bonds with SER-212 and LYS-97 toward MAP2K1, while ARG-40 and ASP60 were built between FDX1 and LANRAPLENIB (Fig 10A–10D).

## Analysis of hub-genes at the single-cell level

To create a gene expression panorama of the 5 hub-genes in the peripheral immunocytes at the single-cell level, we generated scRNA-seq profiles of peripheral blood from 3 patients with AD and 2 healthy controls (GSE181279 dataset). After quality control procedures (S2A Fig), the correlations of sequencing depth were mapped out (S2B Fig). The nCount RNA vs nFeature RNA was 0.92, and the nCount RNA vs percent.mt was −0.01, indicating the good quality of our scRNA data.

A total of 36758 cells were retained to normalize, and then PCA was employed to downscale the dimensionality of the scRNA-seq data (S2C and S2D Fig). The feature genes of each principal component were displayed in S2E and S2F Fig. Furthermore, 23 cell clusters were classified utilizing t-SNE analysis (Fig 11A, S7 Table), and then the cell populations

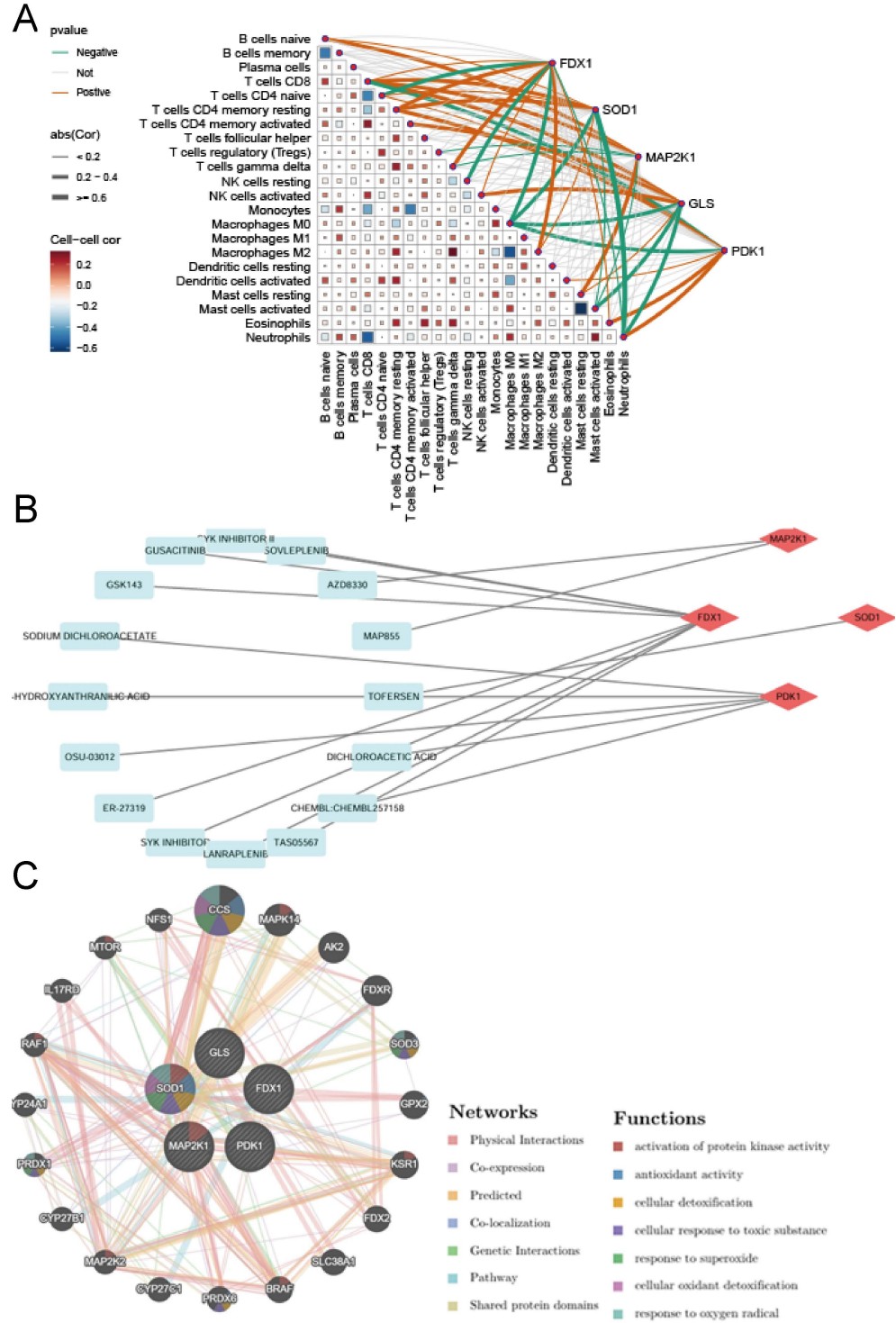

**Fig 9. Characterizations of hub-genes related immune infiltration, drugs, and genes. (A)** Analysis of the relationship between hub-genes and immunocytes, as well as among different immunocytes. **(B)** The interacting network of the hub-genes and targeted drugs. **(C)** The predicting targeted genes using the GeneMANIA database.

**Table 3. Molecular docking analyses of the picked predicted drugs toward target hub-gene related proteins.**

| Predicted dugs | Target hub-gene related proteins | Binding energy (kcal/mol) |
|---|---|---|
| Tetracycline | SOD1 | −7.8 |
| AZD8330 | MAP2K1 | −7.5 |
| LANRAPLENIB | FDX1 | −7.3 |
| Dichloroacetic Acid | PDK-1 | −7.7 |

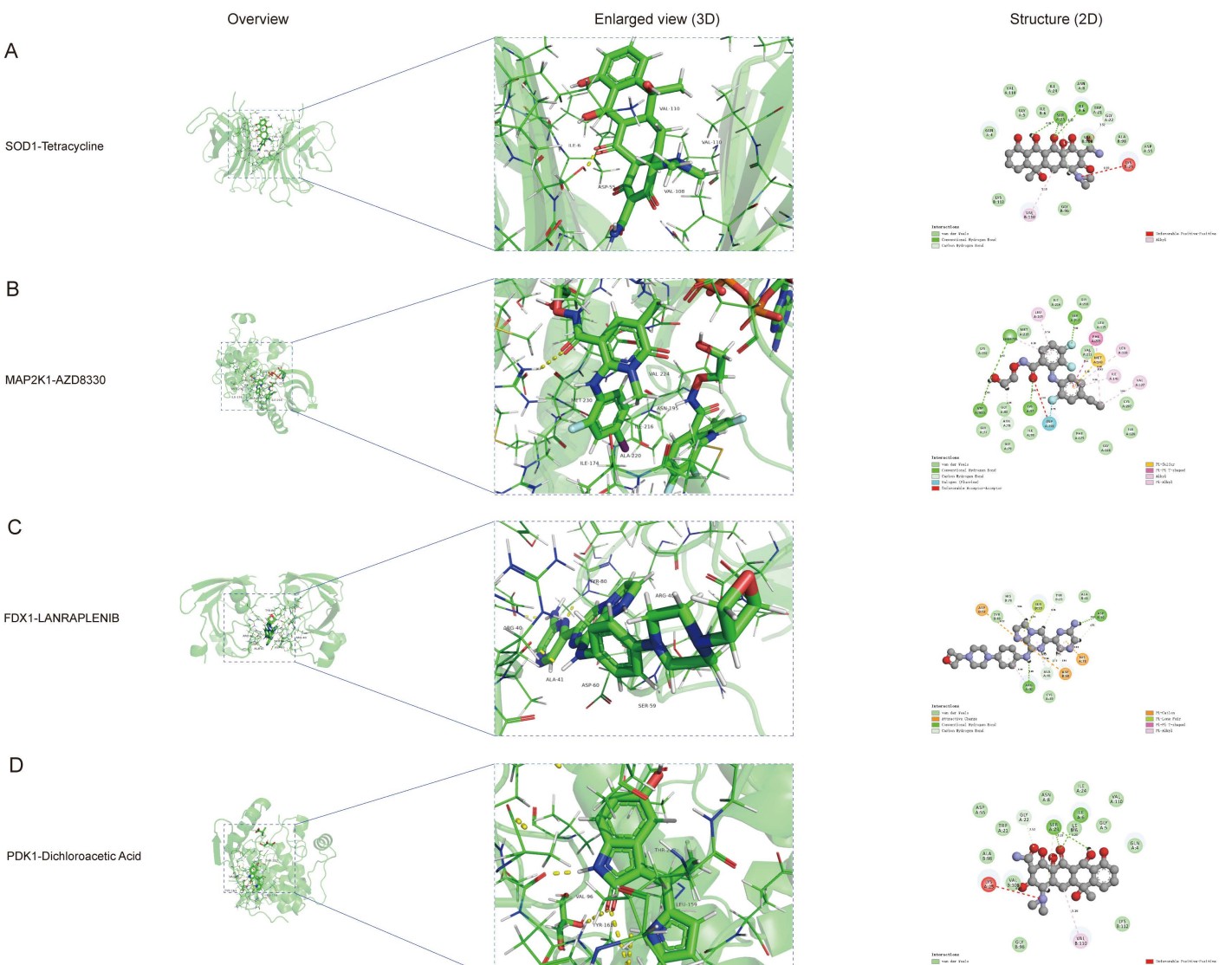

**Fig 10. Molecular docking in each hub-gene related protein with the picked predicted drugs. (A)** SOD1 and Tetracycline; **(B)** MAP2K1 and AZD8330; **(C)** FDX1 and LANRAPLENIB; **(D)** PDK1 and Dichloroacetic Acid.

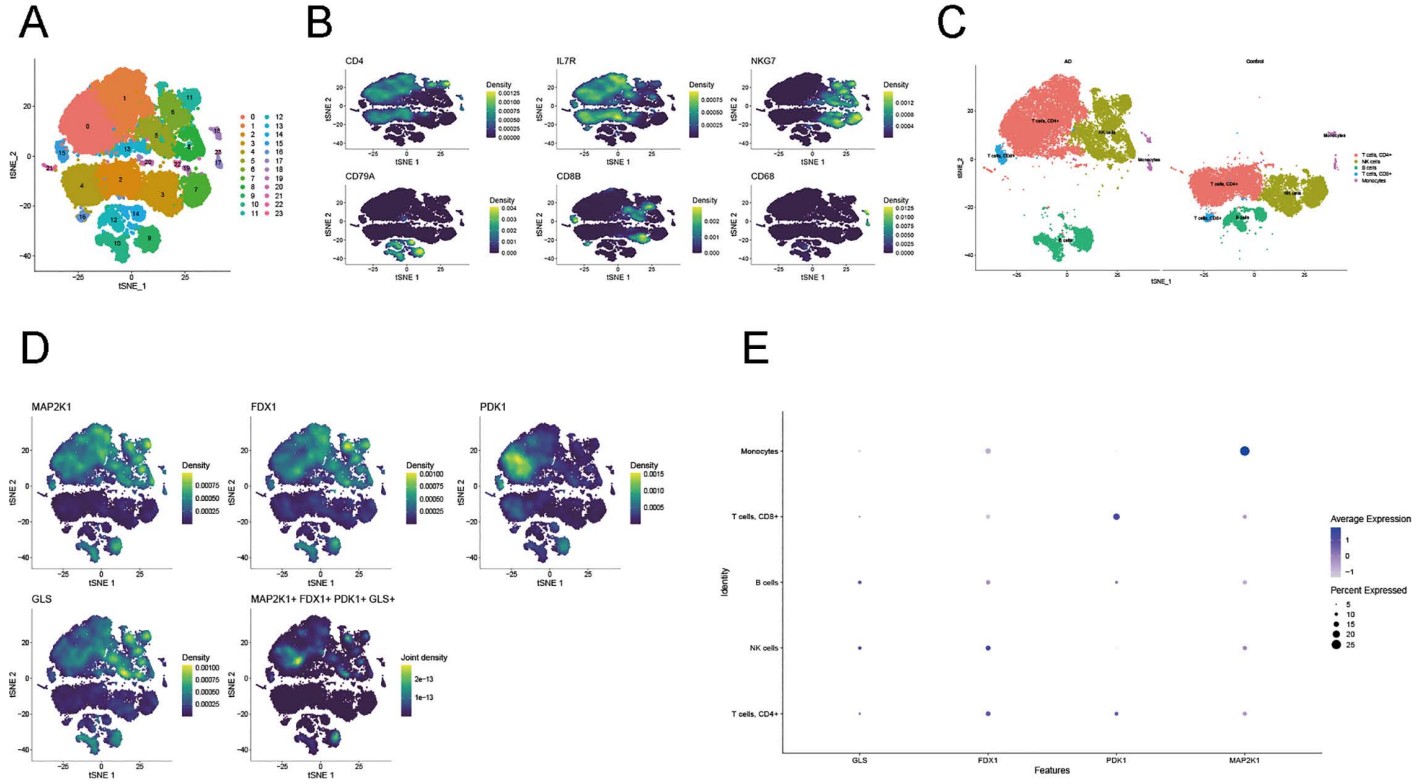

**Fig 11. Cell type classification and analysis of the hub DECAGs at the single-cell level.** (A) A t-SNE plot illustrating cell clustering. (B) t-SNE plots demonstrating typical cell surface markers defining CD4+ T cells (CD4, IL7R), NK cells (NKG7), B cells (CD79A), CD8+ T cells (CD8B), and monocytes (CD68). (C) A t-SNE plot visualizing grouped cell annotation. (D) t-SNE plots demonstrating the distribution of the picked hub genes and their co-expression in cells. (E) A bubble plot demonstrating the selected hub DECAGs for each cell type in AD.

were annotated according to different expressed markers genes (S8 Table). We eventually determined five main immu-nocyte types, including B cells, CD4+T cells, NK cells, CD8+T cells, and monocytes (Fig 11B, S2G and S2H Fig). Subse-quently, we grouped them into AD and healthy control in the t-SNE plot (Fig 11C). Notably, we found out that the AD and the healthy control group were separated in the t-SNE plot, implying that immune cells' gene expression patterns in AD patients may differ.

To elucidate the roles of the 5 hub-genes at the single-cell level, we screened DECAGs for each immune cell type between the AD group and healthy controls and found that except *SOD1*, all of the hub genes were significantly dif-ferentially expressed in a particular immune cell type or multiple immune cell types (S2I Fig). Therefore, we selected *MAP2K1, FDX1, GLS*, and *PDK1* for our next study. We visualized these picked hub DECAGs in the t-SNE plots as well as observed significantly elevated expression of *MAP2K1* and *FDX1* among all five immune cell subsets, while *GLS* was highly expressed in NK cells, CD4+T cells, and B cells. CD8+T cells, CD4+T cells, and B cells showed higher expression levels of *PDK1*. Notably, the visualization of their co-expression revealed the picked hub DECAGs were mainly concen-trated among NK cell, B cell, and CD4+T cell clusters of AD samples (Fig 11D, S2J and S2K Fig). Furthermore, we evalu-ated the expression levels of *MAP2K1, FDX1, GLS*, and *PDK1* of these immunocyte types among AD samples and found that *MAP2K1* and *FDX1* were enriched in all cell populations; *GLS* was abundantly expressed in NK cells, B cells, and CD4+T cells, whereas *PDK1* was highly expressed among B cells, CD8+T cells, and CD4+T cells, which corresponded to the results of S2 Fig and Fig 11D (Fig 11E).

## Intercellular communication characteristics

Cellular interactions between peripheral immunocytes participate in the genesis of many medical conditions, and a growing amount of research indicates that the development of AD is related to peripheral immunity. We therefore established a cell-cell communication framework in the AD group's blood to explore cellular interactions between peripheral immune cells through the association of ligands with the corresponding receptors. Initially, we observed that in aggregated cellular communication networks with roughly the same number of interactions, B cells interacted extremely strongly with all other cells (including themselves) (Fig 12A and 12B). We further analyzed the interacting ligand-receptor pairs among the five immune cell subsets in the AD samples (S9 Table), in which we identified 50 important ligand-receptor pairs distributed in nine signaling pathways, including MIF, GALECTIN, RESISTIN, ANNEXIN, IL16, TNF, PARs, BAFF, and APRIL pathways. We then examined the probability of communication between ligands and receptors for five immune cell types and found that the MIF pathway signaled most strongly between different immune cells (Fig 12C). Therefore, we focused on the MIF signaling pathway and found that CD8+T cells were the main MIF senders, while B cells were the predominant MIF receivers (Fig 12D, 12E, S2L Fig). We further explored cell-cell communication at the MIF pathway level and its two corresponding ligand-receptor pairs level (Fig 12F, S2M Fig). We discovered that MIF − (CD74+CD44) plays a key role in the MIF signaling pathway (S2N Fig), in which B cells and monocytes are the main receivers while B cells are the only receptor cells in the MIF – (CD74+CXCR4) pathway (Fig 12G).

## Clinical evaluations and validation by RT-qPCR

We finally validated the expression profiles of correlated risk genes in blood specimens collected from AD patients and healthy controls in our clinical center. The clinical information for each AD patient is in Fig 13A, including amyloid PET and MMSE scores. Using qPCR detection, we discovered that the levels of *PDK1* expression in AD patients were remarkably upregulated, while *FDX1* and *GLS* were significantly decreased (Fig 13B, S3 Fig).

## Discussion

Recently, the role of cuproptosis functional disorder in AD is noteworthy, and cuproptosis-associated genes (CRGs) have emerged as potential therapeutic targets and diagnostic biomarkers for various neurodegenerative diseases [25]. Nevertheless, the specific molecular mechanisms and precise role of cuproptosis in AD remain obscure. In this study, the DECAGs between AD patients and normal controls were analyzed, combining machine learning methods such as XGB, RF, SVM, and RF models, to screen for diagnostic markers capable of predicting AD risks. Moreover, we have meticulously constructed comprehensive transcriptome profiles of blood from AD patients using scRNA-seq datasets, revealing activated immune functions. Finally, the five genes' expression levels were verified by RT-qPCR experiments, confirming *FDX1, PDK1,* and *MAP2K1* as potential AD-specific markers.

By analyzing RNA sequencing data, we identified 19 differentially expressed cuproptosis-associated genes. Some of these genes have been reported to be directly involved in the pathogenesis of AD. A study finds that lead-induced *COX17*-regulated mitochondrial copper accumulation exacerbates pathology in Alzheimer's disease [50]. *MTF-1* was identified as a key transcription factor that plays an important role in the pathogenesis of cognitive impairment induced by 1,2-diacetylbenzene [51]. Scientists have found that overexpression of metallothionein-1 (*MT-1A* and *MT-1B*) modulates the pathogenesis of AD through antioxidant, anti-inflammatory, and heavy metal binding [52]. *PDHA1* deficiency in the hippocampus results in lactate accumulation that may inhibit the cAMP/PKA/CREB pathway, thereby impairing cognitive function [53]. Inhibition of the core metabolic enzyme dihydrolipoamide dehydrogenase (*DLD*) has been reported to protect against amyloid beta toxicity in AD [54]. These findings suggest a close relationship between the DECAGs and Alzheimer's disease.

Based on the GO and KEGG analyses, we determined the main biological processes and molecular functions of these DECAGs associated with AD. The GO results indicated that these functions were strongly correlated with acetyl-CoA

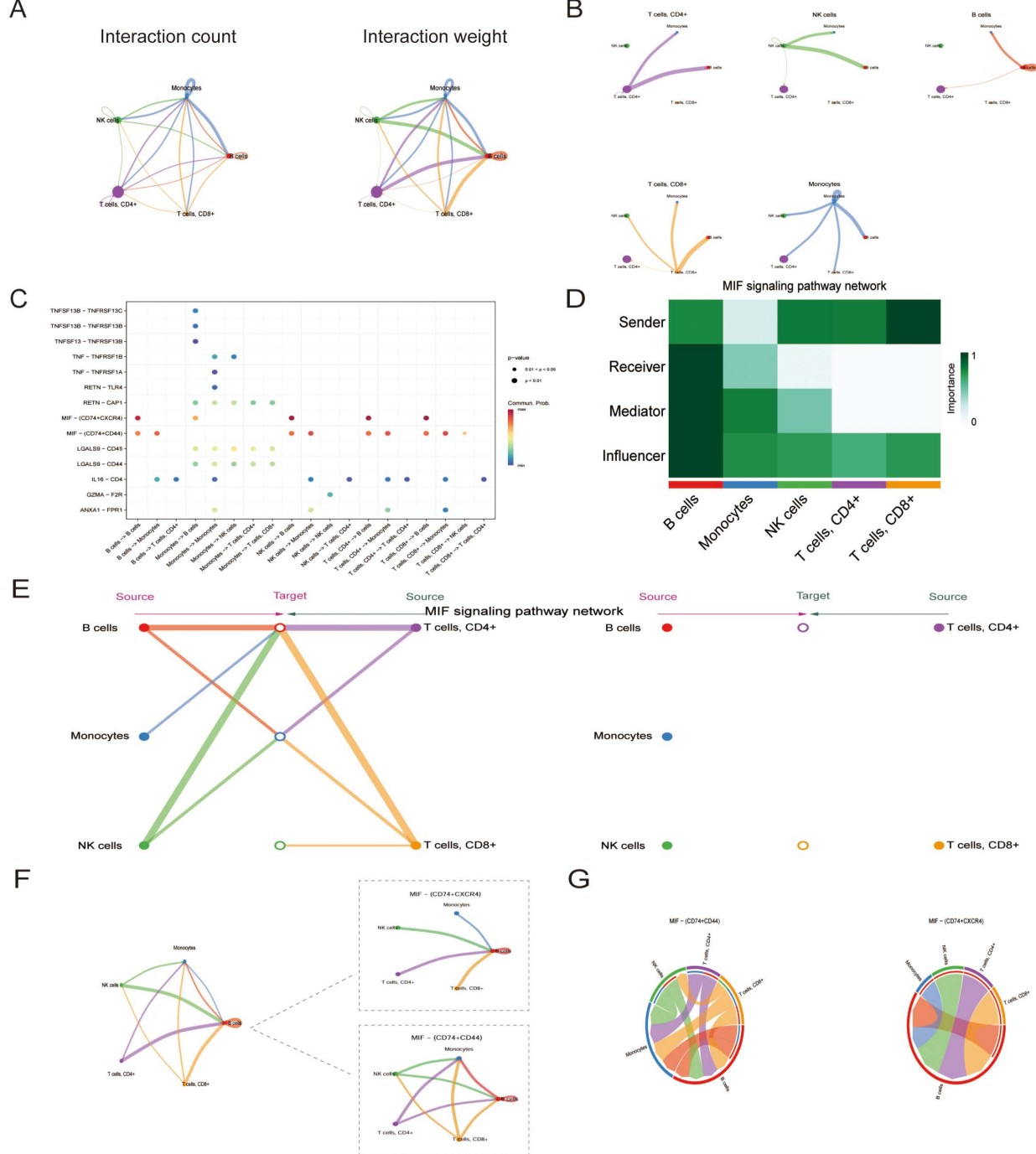

**Fig 12. Intercellular communication in the peripheral immune cells of AD patients.** (A) Circle plots assessing the cell-cell network communications. The thickness of the edges represents the number (left) or the weight (right) of interactions among peripheral immune cell types. (B) Circle plots demonstrating the cell-cell interaction weights among each peripheral immune cell separately. (C) Probability of communication between ligands and receptors in peripheral immune cell populations. Signal enhancement indicates an increase in the communication potential of these signals. (D) A heatmap illustrating the roles of each type of peripheral immune cells in the MIF signaling pathway. (E) Hierarchical analysis of the intercellular communication network of the MIF signaling pathway. The same colors represent the same cells. (F) Circle plots demonstrating cellular communication at the level of the MIF pathway and its corresponding ligand-receptor pairs level. (G) Chord plots demonstrating cellular communication in the MIF signaling pathway at the level of ligand-receptor pairs.

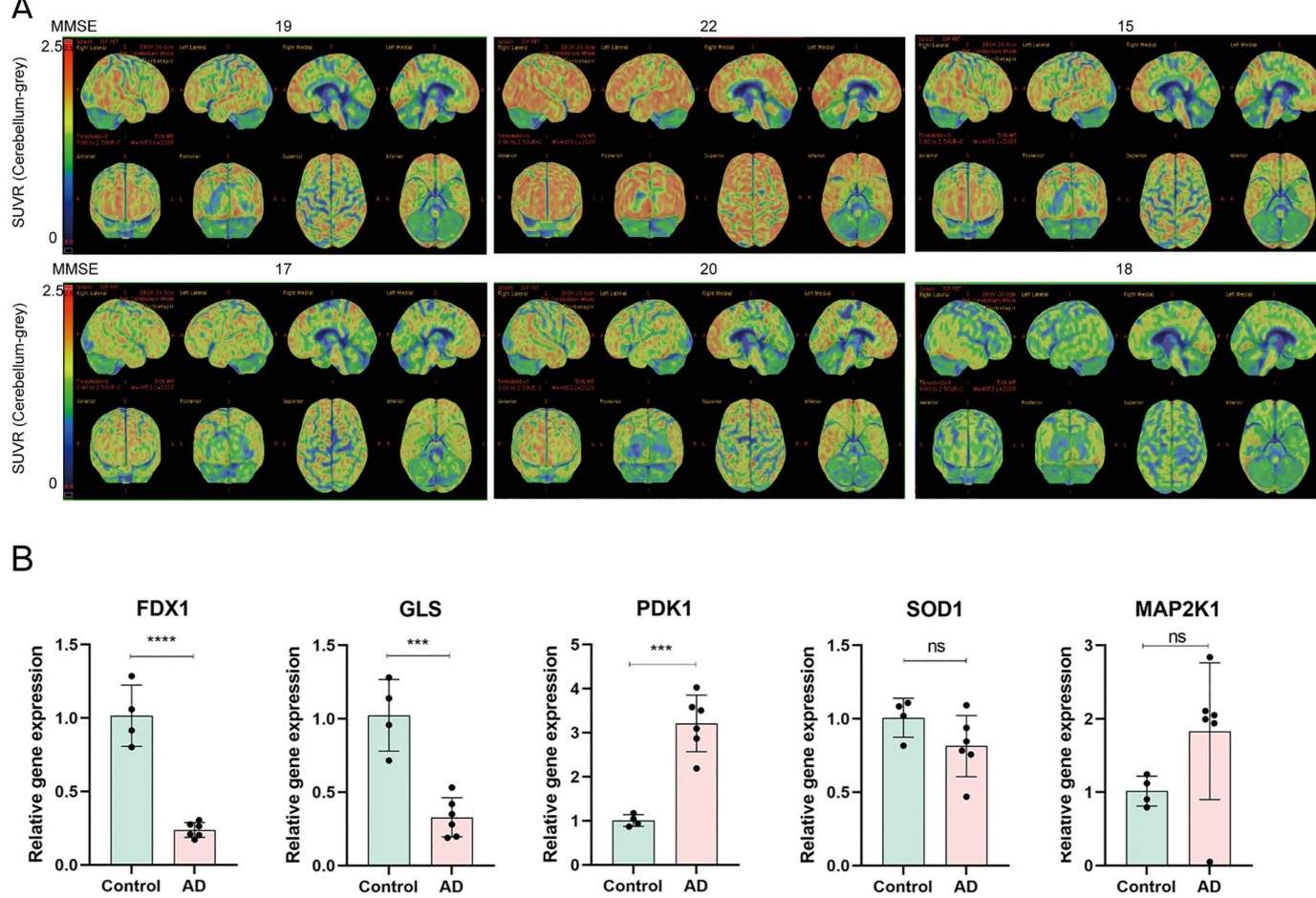

**Fig 13. (A) Topographical plots of amyloid (18F-florbetapir) PET SUVr levels for the 6 AD patients.** (each column represents one participant, named after their baseline MMSE score). (B) The relative gene expression of five hub genes between AD patients and normal controls using GAPDH as internal references.

biosynthetic and metabolic processes, oxidoreductase complex, oxidoreductase activity, and copper ion binding. The KEGG pathways characterized by DECAGs were mainly focused on 2-Oxocarboxylic acid metabolism, Citrate cycle, and lipid acid metabolism. Most of these terms were focused on mitochondrial metabolism, and all have been reported to be closely related to the pathogenesis of AD [55–59]. In addition, the bioinformatics analysis revealed an intimate relationship between DECAGs and immune cells.

Moreover, we determined *MAP2K1, FDX1, GLS*, and *PDK1* as five hub DECAGs predicting AD risks using machine-learning algorithms in the peripheral blood samples. Further, by using GSEA analysis of the 5 hub-genes, we have understood the GO functions and pathways of these hub DECAGs in AD and normal groups, which were not only mainly focused on the cytoplasmic translation, ncRNA processing, and oxidative phosphorylation that were comparable as the results of the GO and KEGG analysis mentioned above, but also on regulation of immune response. Analysis of the relationship between hub-genes and immune cells infiltration demonstrated that all of the five genes were significantly cor-related with various immunocytes. From related literature, ferredoxin 1 (*FDX1*) is a mitochondrial electron transmitter that is essential for the synthesis of lipoic acid, which has been implicated in the regulation of cuproptosis. Its altered expres-sion in AD may lead to mitochondrial dysfunction, which is a feature of the disease [60]. Furthermore, the associations

of *FDX1* with CD4+T cells suggest a link between mitochondrial dysregulation and immune cell dysfunction, highlighting its dual role in neuronal and immune homeostasis [61]. The *GLS* encoding glutaminase regulates glutamate metabolism, which is critical for neuronal function and immune cell activation [62]. Dysregulated glutamate metabolism in AD not only exacerbates excitotoxicity but also affects the activity of NK and B cells, which are dependent on glutamate catabolism for energy and effector functions [63]. This double involvement highlights the potential of glutamate metabolism as a biomarker of neuroinflammation in AD. *MAP2K1*, a key component of the MAPK signaling pathway, is crucial to cell proliferation, survival, and immune responses [64]. In AD patients, its dysregulation may alter CD4+T cell activation and recruitment, leading to neuroinflammation and subsequent cognitive decline. Superoxide dismutase 1 (*SOD1*) protects against oxidative stress, a prominent feature in AD. Its interactions with immune cells, particularly B cells, suggest a role in modulating oxidative stress-induced immune responses. Serine/threonine kinase 3-phosphatidylinositol-dependent protein kinase 1 (*PDK1*) is important for multiple types of immune cell development and function, including T cells, B cells, and NK cells. Scientists have found that it is involved in regulatory T cell survival through controlling redox homeostasis [65]. It may improve AD by targeting Foxp3+regulatory T cells to break immune tolerance [66]. Taken together, we hypothesize that these hub DECAGs genes are involved in the peripheral immune modulation, which may contribute to the pathogenesis of AD.

We explored the immune microenvironment in AD patients through immune infiltrating analyses. It was found that the percentage of immune cells inducing neuroinflammation was increased in AD patients compared to controls, such as NK cells, B cells, T follicular helper cells, CD8 T cells, and macrophages [67–71]. Moreover, there was a decrease in immunocytes that suppress the immune system in AD patients, especially Treg [72]. These results uncovered a pro-inflammatory microenvironment in AD patients. Based on the expression profiles of DECAGs in the AD group, we found that the AD patients could be stably divided into two groups. GSVA analysis of the immune pathways of the two clusters showed distinct immune characteristics of the two clusters. Cluster 1 was closely associated with immunosuppressive pathways, particularly the IL-4 pathway, which may inhibit the progression of neuroinflammation in AD [73]. The GSVA results also indicated that the DECAGs may be involved in the progression of neuroinflammation in AD patients. Subsequently, the correlation analysis of the 5 hub-DECAGs with AD peripheral immune cells verified that these genes may modulate immune function in AD.

We provided further insights into the gene expression panorama of the hub DECAGs in peripheral immunocytes at the single-cell level and found they were predominantly concentrated in NK cell, B cell, and CD4+T cell clusters in AD samples. The analysis of intercellular communication indicated the MIF signaling pathway signaled most strongly between different immune cells and B cells which were the predominant receivers in it. The macrophage migration inhibitory factor (MIF) pathway has a pivotal relationship in shaping the immunological microenvironment in Alzheimer's disease [74]. MIF is a multifactorial cytokine with complex immunomodulatory effects, particularly through interaction with the CD74 and CD44 receptor complexes [75]. This ligand-receptor interaction drives key signaling cascades involved in neuroinflammation and immune cell recruitment, processes that are highly dysregulated in AD. Notably, recent studies have highlighted the unique interplay between MIF and B lymphocytes in AD [76]. B lymphocytes, traditionally thought to have antibody-mediated functions, have now emerged as key regulators of immune homeostasis within the brain. Ascending levels of MIF in the AD brain are known to be linked to increased B-cell infiltration and activation, potentially leading to neuroinflammation [77]. In addition, MIF signaling via CD74 and CD44 on B lymphocytes promotes B lymphocyte survival, migration, and pro-inflammatory cytokine production, thereby exacerbating the chronic inflammatory milieu characteristic of AD. A study has lately reported that cuproptosis-related genes have the potential to affect the MIF pathway, including *FDX1* and *GLS* [78]. Therefore, we speculated that these hub DECAGs correlated immune cell types, might influence the peripheral immunity of AD through the MIF signaling pathway which required further validation.

Predicting the binding affinity of a molecule to a specific biological target is critical for the rational design of AD therapies, contributing to the development of more effective, selective, and safer drugs to combat this complex

neurodegenerative disease [79–81]. Therefore, we performed molecular docking analyses of the highest scoring predicted drugs in the DGIdb database to further validate their AD therapeutic potential, including Tetracycline, AZD8330, Lanraplenib, and Dichloroacetic Acid (DCA). The binding affinity scores were all below 7 kcal/mol, which forecasted multiple strong hydrogen bonding interactions between the four predicted drugs and the target proteins, revealing that these drugs may be intimately interconnected with the corresponding hub-genes. Tetracycline, a broad-spectrum antibiotic, has been studied in recent years and found to have potential therapeutic benefits for AD, including inhibition of Aβ aggregation, anti-inflammatory, antioxidant, and metal ion chelating effects [82,83]. The MEK/ERK pathway is an important intracellular signaling pathway, and aberrant activation of this pathway is closely associated with neuroinflammation, synaptic dysfunction, and neuronal death, which is involved in the progression of AD [84–86]. AZD8330 is a selective MEK1/2 inhibitor [87] and may modulate the hyperactivation of the MEK/ERK signaling pathway, thereby attenuating AD-related pathological processes. Lanraplenib, a selective spleen tyrosine kinase (SYK) inhibitor, was originally developed to treat autoimmune diseases and cancer. Evidence suggests that it can inhibit signaling downstream of B cell receptors and reduce the release of pro-inflammatory cytokines such as TNF-α and IL-1 from human macrophages [88], thereby exerting an anti-inflammatory effect that might ameliorate neuroinflammation in AD. DCA is a small molecule compound, that promotes oxidative glucose metabolism and increases mitochondrial energy production by inhibiting pyruvate dehydrogenase kinase (PDK) and activating pyruvate dehydrogenase complex (PDH) [89]. Moreover, recent research has shown that it can reduce amyloidogenic and increase non-amyloidogenic proteolysis of the amyloid precursor protein [90]. The molecular docking results suggest that the identified compounds have strong predicted binding affinities to the target protein. However, it is important to note the limitations of molecular docking. For example, the target proteins are not flexible enough because molecular docking typically uses rigid protein structures that may not fully capture the dynamic nature of protein- ligand interactions [91,92]. In addition, the scoring functions used to predict binding affinities may not always accurately reflect real-world binding energies [93–95]. These limitations highlight the need for targeted experimental validation to translate our docking results into actual drug effectiveness, including structural biology studies, biophysical assays, cell-based experiments, and pharmacological studies [45,96–98].

In recent years, the field has shifted from diagnosing and characterizing AD according to the manifestations of the syndrome to defining the disease biologically. Early detection of AD remains challenging owing to the diversity of clinical subtypes and their heterogeneity [4]. Molecular biomarkers including PET, CSF, and plasma, offer a valuable complement to MRI and clinical assessments in AD diagnosis [99]. However, PET is not available in most of the clinical institutions and cerebrospinal fluid (CSF) biomarkers have shown promise in several studies though, their collection is often invasive and costly. In contrast, plasma biomarkers present a safer, minimally invasive alternative, providing insight into the biological state of the organism. Thus, the identification of biomarkers from peripheral blood is of great clinical value and provides a simpler and easier method for AD diagnosis. We performed biological validation using blood specimens collected from biologically confirmed AD patients and matched healthy controls and found significant changes in the expression of three genes: *FDX1, GLS*, and *PDK1*, implicating that these genes could be used as easily available biomarkers for predicting AD occurrence.

However, there are several limitations in this study that should be noted. First, cuproptosis, as a newly defined programmed cell death, has already been reported in several studies to be strongly associated with Alzheimer's disease. Yongxing Lai et al. used bulk mRNA brain tissue datasets from AD patients to identify two stable AD subtypes based on CAGs and constructed a promising AD prediction model to assess their risk [100]. Bin Nie et al. used blood bulk mRNA from AD patients to classify two clusters according to CAGs profiles, indicating that cuproptosis indeed plays an important role in AD subtypes, and further used WGCNA and drug prediction methods to identify therapeutic biomarkers of AD [101]. All of these studies have performed immune infiltration analyses, suggesting that CAGs closely correlate with neuroinflammation in AD. In our study, we used larger numbers of CAGs to identify biomarkers in AD peripheral immune systems, and the promising results from two external datasets of AD brain tissue validated their universality. Notably, we uniquely linked

the biomarkers to peripheral immune dysregulation and immune cell communication in AD via single-cell RNA-seq and CellChat analysis. These provided a stronger, more targeted hypothesis regarding cuproptosis involved in AD progression. Unfortunately, all of these studies are based on bioinformatics analyses. It is imperative to further investigate the mechanisms of cuproptosis. Second, although our study provided a strong relationship between CAGs and AD in the peripheral immune system using multi-omics studies, the bulk and single-cell RNA data were obtained from a limited number of samples. Third, we discovered immune alterations in AD peripheral blood, but the specific mechanisms between cuproptosis and clonal alterations in immune cells remain unknown. Fourth, we were unable to complete overlap analyses of blood with CSF or cerebral tissue due to the lack of adequate samples. More detailed studies on how these genes (e.g., *FDX1*, *PDK1*, *MAP2K1*) regulate cuproptosis are needed to further delineate the intrinsic AD mechanisms.

## Conclusion

In conclusion, our study supported therapeutic strategies based on cuproptosis genes as promising immune mediators. Machine-learning algorithms identified five hub cuproptosis genes involved in AD progression. Single-cell analyses help provide insight into the molecular mechanics of peripheral immune dysregulation of AD. The results offer novel targets for AD intervention and provide insights into the mechanisms underlying AD pathogenesis.

## Supporting information

**S1 Fig. Gene set enrichment analysis for the five genes in AD patients.** KEGG pathway analysis revealed significant biological processes in the high and low expressions of FDX1 (A), MAP2K1 (B), PDK1 (C), GLS (D), and SOD1 (E). (DOCX)

**S2 Fig. Single-cell analysis and cell-cell communication.** (A) Quality control of the GSE181279 dataset. (B) Scatter plot of the correlations of sequencing depth. (C) Elbow plot of the principal components in PCA. (D) PCA plot showing the downscale results. (E) The feature genes of each principal component. (F) Heatmap of the PCA feature genes. (G) A t-SNE plot visualizing cell annotation. (H) A bubble plot demonstrating typical cell surface markers defining five immune cell types, including CD4+T cells, NK cells, B cells, CD8+T cells, and monocytes. (I) Volcano plots of DEGs in five immune cell types between AD patients and healthy controls. (J) Scatter plots demonstrating the distribution of hub-genes in cells of the AD and Control groups. (K) A bubble plot demonstrating the percentage expression of hub-genes in each immune cell in the AD and Control groups. (L) A heatmap illustrating the MIF signaling network. (M) A violin plot demonstrating the gene expression levels of the MIF signaling pathway. (N) The contribution of the MIF signaling pathway. (DOCX)

**S3 Fig. Amplification and lysis curves for each tested gene.** (A) *SOD1*. (B) *FDX1*. (C) *PDK1*. (D) *GLS*. (E) *MAP2K1*. (F) *GAPDH*. (DOCX)

**S1 Table. The specific information of datasets in this study.** (DOCX)

**S2 Table. The 46 cuproptosis-related genes screened from available literature.** (XLSX)

**S3 Table. The enriched GO terms of the 19 DECAGs.** (XLSX)

**S4 Table. The significant enriched KEGG terms of the 19 DECAGs in AD.** (XLSX)

**S5 Table. The results of the consensus clustering based on the DECAGs expression profiles.**
(XLSX)

**S6 Table. Interaction scores of the candidate drugs for each hub-gene.**
(XLSX)

**S7 Table. Identification of differently expressed genes in each cell cluster.**
(XLSX)

**S8 Table. Identification of differently expressed genes in every cell.**
(XLSX)

**S9 Table. Intercellular communication networks in the AD samples.**
(XLSX)

## Acknowledgments

We appreciate the public databases GO, KEGG, and Metascape for their online resources.

## Author contributions

**Conceptualization:** Zi-Wen Yu, Nai-An Xiao, Kun-Mu Zheng, Bin Jiang.

**Data curation:** Jing Wang, Zi-Wen Yu, Qi Liu.

**Formal analysis:** Jing Wang, Qi Liu.

**Investigation:** Hui-Juan Wan.

**Methodology:** Jing-Xun Wu, Yi-Dan Zhang.

**Resources:** Min Bi.

**Writing – original draft:** Jing Wang, Zi-Wen Yu, Qi Liu.

**Writing – review & editing:** Nai-An Xiao, Kun-Mu Zheng, Bin Jiang.

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
