## [Decision Letter · Decision Letter 0]

18 Feb 2025

PONE-D-25-01107Novel insights from integrated analysis: the role of cuproptosis and peripheral immune infiltration in Alzheimer's diseasePLOS ONE

Dear Dr. Wang,

Thank you for submitting your manuscript to PLOS ONE. After careful consideration, we feel that it has merit but does not fully meet PLOS ONE’s publication criteria as it currently stands. Therefore, we invite you to submit a revised version of the manuscript that addresses the points raised during the review process.

We look forward to receiving your revised manuscript.

Kind regards,

Li Shen

Academic Editor

PLOS ONE

Journal Requirements:

“This work was supported by grants from the Natural Science Foundation of Fujian Province to Bin Jiang (No.2022J011359) and Jing-Xun Wu (No.2022J011380), the Medical and Health Guiding Project of Xiamen City to Qi Liu (No.3502Z20224ZD1013) and Jing-Xun Wu (No.3502Z20214ZD1011).”

6. PLOS requires an ORCID iD for the corresponding author in Editorial Manager on papers submitted after December 6th, 2016. Please ensure that you have an ORCID iD and that it is validated in Editorial Manager. To do this, go to ‘Update my Information’ (in the upper left-hand corner of the main menu), and click on the Fetch/Validate link next to the ORCID field. This will take you to the ORCID site and allow you to create a new iD or authenticate a pre-existing iD in Editorial Manager.

7. Your ethics statement should only appear in the Methods section of your manuscript. If your ethics statement is written in any section besides the Methods, please move it to the Methods section and delete it from any other section. Please ensure that your ethics statement is included in your manuscript, as the ethics statement entered into the online submission form will not be published alongside your manuscript.

Reviewers' comments:

Reviewer's Responses to Questions

**Comments to the Author**

1. Is the manuscript technically sound, and do the data support the conclusions?

Reviewer #1: Partly

Reviewer #2: Yes

Reviewer #3: Yes

Reviewer #4: Yes

2. Has the statistical analysis been performed appropriately and rigorously? 

Reviewer #1: No

Reviewer #2: I Don't Know

Reviewer #3: I Don't Know

Reviewer #4: Yes

3. Have the authors made all data underlying the findings in their manuscript fully available?

Reviewer #1: Yes

Reviewer #2: No

Reviewer #3: Yes

Reviewer #4: Yes

4. Is the manuscript presented in an intelligible fashion and written in standard English?

Reviewer #1: Yes

Reviewer #2: Yes

Reviewer #3: Yes

Reviewer #4: Yes

5. Review Comments to the Author

Reviewer #1: Dear Authors,

I have found some similar studies: https://www.frontiersin.org/journals/molecular-biosciences/articles/10.3389/fmolb.2024.1478611/full. which closely resembles the figure type presented in this manuscript. Additionally, other related studies include:

1. https://www.frontiersin.org/journals/aging-neuroscience/articles/10.3389/fnagi.2022.932676/full

2. https://www.frontiersin.org/journals/aging-neuroscience/articles/10.3389/fnagi.2023.1204530/full

Moreover, I noticed that the dataset and methodology used in this manuscript appear to be the same as in previous studies. The authors should establish a strong hypothesis and present a novel approach to distinguish their work.

• The authors have utilized array datasets rather than sequencing. Since sequencing offers an advantage over microarrays in detecting novel transcripts, was sequencing data unavailable?

• PCA was conducted on DECAGs, with only two clusters chosen for further analysis. Were these clusters selected based on a specific p-value or eigenvectors with the highest eigenvalues? Additionally, please specify the number of data points mapped to these two clusters.

• For the machine learning models, 70% of the dataset was used for training, while 30% was reserved for testing. However, the authors have not mentioned the number of epochs or the cross-validation techniques applied. Furthermore, the dataset should ideally be divided into training, validation, and testing subsets. Please discuss.

Thanks

Reviewer #2: This manuscript presents an engaging and potentially impactful study exploring the role of cuproptosis-related genes in Alzheimer's disease (AD). However, several issues need to be addressed in order to improve the clarity of gene selection, the statistical validation of machine learning models, and the reconciliation of discrepancies between different experimental techniques.

1, The machine learning models were used to identify hub-genes, but the statistical significance of these findings is not adequately demonstrated. It remains unclear whether cross-validation was performed and how the performance of each model was evaluated. The manuscript mentions that XGB automatically selects the best features based on weights, but the criteria or thresholds used for feature selection are not specified. Although the manuscript provides general descriptions of the machine learning algorithms, a more detailed explanation of the specific configurations, parameters, and any custom modifications to the standard algorithms would be beneficial. For instance, how were the parameters for RF and SVM optimized? Furthermore, the manuscript mentions several R packages (e.g., "caret", "randomForest", "kernlab", "xgboost", "e1071", "rms"), which is useful for reproducibility, but the version numbers of these packages should be provided to avoid discrepancies in future replication attempts. Lastly, a detailed explanation of how points were assigned to each gene in the nomogram, as well as the calculation of total points, would be advantageous. The statistical methods used to assign these points should also be clearly outlined.

2, Molecular docking is a widely used and effective tool for investigating protein-ligand interactions and predicting binding affinities. While the manuscript mentions selecting compounds with the highest docking scores for further analysis, the criteria for selecting these compounds are not clearly defined. Was drug selection based solely on docking scores, or were other factors, such as drug similarity, solubility, and bioavailability, considered? A more detailed explanation of the selection process would improve the transparency of the study.

The manuscript reports that the binding energies of the selected compounds with their respective hub-gene related proteins are all below -7.0 kcal/mol, suggesting strong binding affinity. However, more context regarding this threshold would be helpful. Is this value supported by the literature, or is it specific to this study?

While the docking results provide valuable structural insights, the biological relevance of these findings in the context of Alzheimer's disease (AD) is not fully addressed. How do the predicted binding affinities relate to AD treatment? Are there experimental studies or existing literature supporting the involvement of these proteins and interactions in AD? A deeper discussion of the biological significance of these results would enhance the manuscript's impact.

Reviewer #3: The authors attempt to describe cuproptosis genes and how they might be used as promising immune mediators. They perform a great deal of analytics to accomplish this. Unfortunately, there seems to be somewhat of a disconnect as to what is being done and why for these purposes. The authors describe each process used, but what is lacking is an overall big picture of why each step is being done and what value it adds overall. Clarification would be appreciated to get through the results sections. Maybe figure illustrating all the specific methods/functions/packages and how they relate to each other would be helpful.

As for the methods, numerous packages are mentioned as being used. What would be additionally helpful would be to know exactly which functions without those packages helped produce the analysis. Were the defaults used for each of the functions or where some of the parameters modified.

For the ML methods, it would be nice to clarify what the abbreviations stand for before using them (XGB, SVM, GLM, and RF). It is also unclear if any training/testing datasets were used. Information is lacking as to model performance and how that was evaluated.

For construction and validation of a risk model, it is unclear what the outcome is and what were the dependent variables. How many samples were used here? Overall, this info (sample size) is lacking for much of the analyses mentioned.

Reviewer #4: 1. The title says, "integrated analysis," but the study only uses one dataset (GSE63060) for training. I suggest changing the title unless multiple datasets or data types are combined for analysis.

2. The list of DECAGs like COX17, ATP7A, MTF1, etc., is mentioned, but their role in Alzheimer's Disease (AD) isn’t explained well. It would help to discuss how these genes are known to be involved in AD.

3. The enrichment analyses (GO, KEGG) show an overlap with well-known metabolic pathways. It would be useful to explain whether these pathways are already well-known in AD or if the study provides new insights.

4. The correlation between immune cell infiltration and DECAG expression is interesting, but the manuscript doesn’t fully discuss the impact of immune cell differences between AD patients and healthy controls. It would be useful to explore how these immune changes might affect AD progression.

5. The molecular docking results are intriguing, but the limitations and assumptions behind the method aren’t discussed. It would be good to mention how the docking results might translate to actual drug effectiveness and what experiments would be needed to confirm these findings.

6. The study uses several machine learning algorithms (XGB, GLM, RF, SVM) to predict outcomes, but there is no clear explanation on how the models were tuned or selected. Also, it’s unclear whether any validation methods like cross-validation were used to check the model's reliability.

7. The manuscript mentions using "Seurat" for scRNA-seq analysis, but it doesn’t explain how data quality was checked or how noise was reduced, beyond general normalization and batch correction. More details would be helpful.

8. The manuscript states that the five hub genes were identified using machine learning (ML), but this seems to be incorrect. The hub genes were actually identified by analyzing the protein-protein interaction (PPI) network, specifically using degree analysis to find genes with the most interactions. I suggest changing Figure 1 and the methods section to make it clear that the hub genes were found using the PPI network and degree analysis, not ML.

6. PLOS authors have the option to publish the peer review history of their article (what does this mean? ). If published, this will include your full peer review and any attached files.

**Do you want your identity to be public for this peer review?** For information about this choice, including consent withdrawal, please see our Privacy Policy .

Reviewer #1: No

Reviewer #2: No

Reviewer #3: No

Reviewer #4: No

---

## [Author Response · Author response to Decision Letter 1]

17 Mar 2025

Dear Editor and Reviewer,

On behalf of my co-authors, we are very grateful to you for giving us the opportunity to revise our manuscript, thank you very much for your positive and constructive comments on the manuscript. All journal requests were taken seriously, including that we have removed the Funding section in our manuscript and we have stated the work funders took in the Author’s contributions section (Lines 690-695). And the Ethics approval and consent to participate section has been removed and the related content has been moved to Lines 277-280 in the Methods section.

After carefully reading your comments and suggestions, we have revised and responded according to them and displayed them below:

Reviewer #1

Q1. I have found some similar studies: https://www.frontiersin.org/journals/molecular-biosciences/articles/10.3389/fmolb.2024.1478611/full. which closely resembles the figure type presented in this manuscript. Additionally, other related studies include:

1. https://www.frontiersin.org/journals/aging-neuroscience/articles/10.3389/fnagi.2022.932676/full

2. https://www.frontiersin.org/journals/aging-neuroscience/articles/10.3389/fnagi.2023.1204530/full

Moreover, I noticed that the dataset and methodology used in this manuscript appear to be the same as in previous studies. The authors should establish a strong hypothesis and present a novel approach to distinguish their work.

Response: Thank you for bringing these related studies to our attention and for the constructive feedback regarding the similarity. We would like to address these points as follows:

Similarity in Figure Formats: While our figure design may appear similar to those in the referenced studies, the chosen format was selected to clearly and effectively communicate our specific findings. We think that standard visual representations help facilitate comparisons across studies. Moreover, the content in these pictures is completely different from other studies, including the enrichment analysis, GSVA analysis in the two Clusters, machine-learning models, single-cell analysis, cellular communication, and Pet-CT results.

Dataset and Methodological Overlap: We acknowledge that our dataset and certain methodological elements share similarities with prior work in this area. However, our study differs in several important respects: a. Unlike previous studies, we have incorporated single-cell analysis to a stronger, more targeted hypothesis regarding cuproptosis involved in AD progression. b. The number of CAGs included in this study was greater than other studies, and we employed four machine-learning methods to develop a cuproptosis-related model, which was distinguished from other work. c. Our interpretation of the results diverged from that in the referenced studies. We provided a detailed discussion on how our findings contribute to a deeper understanding of MIF pathway and intracellular communication and the potential implications for future research. We have enhanced our methodological description to emphasize the novel aspects of our approach and how these innovations contribute to a more robust understanding of the underlying biological questions. Please refer to the revised content of Methods, Results, and Discussions in the manuscript.

Q2. The authors have utilized array datasets rather than sequencing. Since sequencing offers an advantage over microarrays in detecting novel transcripts, was sequencing data unavailable?

Response: Thank you very much for your significant question. We acknowledge that RNA-seq offers distinct advantages, particularly in its ability to detect novel transcripts. However, our primary objective was to analyze expression patterns of cuproptosis-associated genes across a large, clinically robust cohort. The selected array datasets provided extensive metadata, high reproducibility, and sufficient coverage of established transcripts, making them ideally suited to our study’s focus on differential expression rather than novel transcript discovery. The microarray datasets available to us were generated under highly standardized conditions, ensuring uniformity across samples. This consistency, coupled with a substantial sample size, allowed us to perform rigorous statistical analyses.

Moreover, we employed a high-throughput sequencing dataset, namely the GSE181279 scRNA-seq dataset, to validate our results in single-cell levels.

Q3. PCA was conducted on DECAGs, with only two clusters chosen for further analysis. Were these clusters selected based on a specific p-value or eigenvectors with the highest eigenvalues? Additionally, please specify the number of data points mapped to these two clusters.

Response: We sincerely appreciate the valuable feedback. We are sorry for our negligence. We have reintroduced the optimum cluster selection based on the consensus matrices and the cumulative distribution function (CDF) curve (Lines 173-174). The results are shown in Figures 5A- C, which indicate that the optimum number is k=2. PCA is performed to intuitively visualize the optimum clustering results (Lines 174-176). Figure 5D shows the points of the scatterplot denoting each sample of the two main components of the cluster (PC1 and PC2), which suggests the clustering analysis when k=2 is reasonable and stable. The number of data points is 90 (Cluster1) and 55 (Cluster2), respectively (Table S5, Lines 357-358). We hope these revisions adequately address your concerns.

Q4. For the machine learning models, 70% of the dataset was used for training, while 30% was reserved for testing. However, the authors have not mentioned the number of epochs or the cross-validation techniques applied. Furthermore, the dataset should ideally be divided into training, validation, and testing subsets. Please discuss.

Response: Thank you very much for pointing this out. We deeply apologize for this omission. We have added that we employed k-fold cross-validation (k=5) to ensure optimal selection in machine learning (Lines 200-201).

We have also added the process of dividing the AD patients in the GSE63060 dataset into a training cohort and a test cohort in Lines 185-186.

We validated the risk model using ROC curves on external datasets (GSE33000 and GSE122063). We have added the validation method in Lines 211-214 and revised the validation results in Lines 397-402.

We hope these revisions and clarifications address your concerns and illustrate the distinct contributions of our study. Thank you again for the opportunity to improve our manuscript.

Reviewer #2

Q1. The machine learning models were used to identify hub-genes, but the statistical significance of these findings is not adequately demonstrated. It remains unclear whether cross-validation was performed and how the performance of each model was evaluated. The manuscript mentions that XGB automatically selects the best features based on weights, but the criteria or thresholds used for feature selection are not specified. Although the manuscript provides general descriptions of the machine learning algorithms, a more detailed explanation of the specific configurations, parameters, and any custom modifications to the standard algorithms would be beneficial. For instance, how were the parameters for RF and SVM optimized? Furthermore, the manuscript mentions several R packages (e.g., "caret", "randomForest", "kernlab", "xgboost", "e1071", "rms"), which is useful for reproducibility, but the version numbers of these packages should be provided to avoid discrepancies in future replication attempts. Lastly, a detailed explanation of how points were assigned to each gene in the nomogram, as well as the calculation of total points, would be advantageous. The statistical methods used to assign these points should also be clearly outlined.

Response: We sincerely appreciate the constructive suggestions. We sincerely apologize for our negligence. In the revised manuscript, we have expanded the Methods section to provide additional details as follows:

Cross-Validation and Performance Evaluation:

We reintroduce the cross-validation methodology in Lines 200-201, stating that the optimal choice of the four machine learning algorithms is derived from 5-fold cross-validation with default parameters. Model performance was assessed using residual analysis, ROC curves (via the “pROC” package), and the “model_performance” function, ensuring that the statistical significance of our findings was adequately demonstrated (Lines 201-204).

Feature Selection in XGB:

The XGB model automatically selects features based on weight built-in algorithm (Lines 200-201), the Gradient Boosting Decision Tree (GBDT). Default criteria embedded within the xgboost package (version 1.7.8.1) were used to determine the optimal number of features. The feature importance was then further assessed using the “variable_importance” function to identify hub genes.

Parameter Optimization for RF and SVM:

By default, the maximum depth of the tree is 6 (max_depth = 6), the learning rate is 0.3 (eta = 0.3), the L2 regularization parameter is 1 (lambda = 1), the L1 regularization parameter is 0 (alpha = 0), and the number of iterations is 100 (nrounds = 100). All of the four models are calculated using built-in algorithms (Lines 196-199). Here, we list some important default parameters of these models as follows: (In the RF model, the default number of trees is 500 (ntree = 500) and the default minimum sample size of leaf nodes is 1 (nodesize = 1). In the SVM model, default regularization parameter is 1(C = 1). We used the “kernlab” (version 0.9.33) and “e1071” (version 1.7.16) packages to minimize classification errors using cross-validation. The models were run with default parameters, which were automatically tuned to find the best classification boundaries, and again 5-fold cross-validation was used to ensure model generalization capabilities. No additional customizations were made to this section.

Overall, all models were configured with default parameters based on 5-fold cross-validation, and no special customization was made. This ensures standardization of model training and facilitates repeated validation of the results.

R packages versions:

We added the version numbers in the “Construction of cuproptosis-related AD risk models” section. Please refer to Lines 198-199 and 208.

Nomogram and statistical methods:

We have added “Each of the five genes was given a different point value, and the points were summed to give the total points. If the total points did not exceed 140, the AD risk was less than 0.1, and if the total points exceeded 240, the AD risk was greater than 0.9.“ Please review Lines 391-394.

Thank you once again for your thoughtful review.

Q2. As for molecular docking, a more detailed explanation of the selection process would improve the transparency of the study.

Response: Thank you for your constructive suggestion regarding the selection of the predicted drugs. We apologize for the ambiguity of the article due to our negligence. Criteria for drug selection were based on interaction scores in gene-drug interaction networks constructed using the DGIdb database and available literature support rather than docking scores. DGIdb is a database that integrates drug-gene interaction data from multiple sources (e.g., rugBank, PharmGKB, TTD, and ChEMBL) and provides comprehensive scores for drugs based on experimental evidence, literature support, bioavailability, binding affinity, and more. We have now added the interaction scores of the candidate drugs for each hub-gene (Table S6, Line 427). Furthermore, we have included relevant references that support the selection of drug candidates based on interaction scores, as well as supporting literature. Please see Lines 230-232 and references 36-40 in the manuscript. We appreciate your attention to detail.

Q3. The manuscript reports that the binding energies of the selected compounds with their respective hub-gene related proteins are all below -7.0 kcal/mol, suggesting strong binding affinity. However, more context regarding this threshold would be helpful. Is this value supported by the literature, or is it specific to this study?

Response: Thank you for your valuable comment. We have now added more relevant references in support of this threshold that indicate strong binding affinity between ligand and receptor. Please see Line 440, references 45-49 in the manuscript. We appreciate your attention to detail.

Q4. While the docking results provide valuable structural insights, the biological relevance of these findings in the context of Alzheimer's disease (AD) is not fully addressed. How do the predicted binding affinities relate to AD treatment? Are there experimental studies or existing literature supporting the involvement of these proteins and interactions in AD? A deeper discussion of the biological significance of these results would enhance the manuscript's impact.

Response: Thank you for your valuable feedback, which suggests a deeper discussion of the molecular docking results and their relationship to AD treatment. We appreciate your insights and have revised the discussion section to be more comprehensive.

In our revised manuscript, we found literature supporting that the predicted binding affinity of a molecule to a specific biological target is critical for the development of AD therapeutics. Also, we concluded the reported possible relevance of the four compounds to AD treatment. For instance, aberrant activation of the MEK/ERK pathway is closely associated with the progression of AD. AZD8330, a newly discovered MEK1/2 inhibitor, may improve AD progression by blocking the MEK/ERK pathway, although direct evidence is lacking.

Please refer to the Lines 621-657 in the revised manuscript. We believe that these revisions significantly enhance the discussion and provide a more thorough context for our findings. Thank you once again for your thoughtful review.

Reviewer #3:

Q1. Maybe figure illustrating all the specific methods/functions/packages and how they relate to each other would be helpful.

Response: Thank you very much for valuable comments, which will improve the quality of our manuscript greatly. We have added the corresponding parts in Figure 1 to illustrate the methods and relationship in our study.

Q2. As for the methods, numerous packages are mentioned as being used. What would be additionally helpful would be to know exactly which functions without those packages helped produce the analysis. Were the defaults used for each of the functions or where some of the parameters modified.

Response: Thank you for your valuable comments. As for unsupervised clustering, we have added details of the modified parameters for the “ConsensusClusterPlus” package in Lines 168-172. For machine learning, we reintroduce the cross-validation methodology in Lines 200-201, stating that the optimal choice of the four machine learning algorithms is derived from 5-fold cross-validation with default parameters. All analyses were performed with default parameters based on 5-fold cross-validation, and no special customization was made. For the single-cell analysis, we have added the modified parameters of PCA. Please see Lines 249-256. For the cell-cell communication and signaling pathway analysis, we have added that all calculations are performed automatically with default parameters. Please see Lines 267-271. Additionally, we have highlighted the defaults or the modified parameters we have already mentioned in each section of the Methods in yellow.

We hope our revision will address your concern.

Q3. For the ML methods, it would be nice to clarify what the abbreviations stand for before using them (XGB, SVM, GLM, and RF). It is also unclear if any training/testing datasets were used. Information is lacking as to model performance and how that was evaluated.

Response: Thank you for your valuable comments. We apologize for the ambiguity of the article due to our lack of consideration. We used to mention the full name of the four machine learning methods in Lines 103-105. We have now added the full name of the abbreviation in the ML methods section to make it more understandable. Please see Lines 187-189. As for the lack of information on the training/testing datasets, we sincerely apologize for our ove

---

## [Decision Letter · Decision Letter 1]

4 Apr 2025

PONE-D-25-01107R1Novel insights from integrated analysis: the role of cuproptosis and peripheral immune infiltration in Alzheimer's diseasePLOS ONE

Dear Dr. Wang,

Thank you for submitting your manuscript to PLOS ONE. After careful consideration, we feel that it has merit but does not fully meet PLOS ONE’s publication criteria as it currently stands. Therefore, we invite you to submit a revised version of the manuscript that addresses the points raised during the review process.

In particular, please specify the following points in your future revision:

Please revise the title to reflect the actual approach according to the reviewer's suggestion.Please provide a more direct comparison with the following related studies to clarify the novelty and contributions of your work:
https://doi.org/10.3389/fnagi.2022.932676

https://doi.org/10.3389/fnagi.2023.1204530

We look forward to receiving your revised manuscript.

Kind regards,

Li Shen

Academic Editor

PLOS ONE

Reviewers' comments:

Reviewer's Responses to Questions

**Comments to the Author**

1. If the authors have adequately addressed your comments raised in a previous round of review and you feel that this manuscript is now acceptable for publication, you may indicate that here to bypass the “Comments to the Author” section, enter your conflict of interest statement in the “Confidential to Editor” section, and submit your "Accept" recommendation.

Reviewer #1: All comments have been addressed

Reviewer #2: All comments have been addressed

Reviewer #3: All comments have been addressed

Reviewer #4: (No Response)

2. Is the manuscript technically sound, and do the data support the conclusions?

Reviewer #1: Yes

Reviewer #2: Yes

Reviewer #3: (No Response)

Reviewer #4: Yes

3. Has the statistical analysis been performed appropriately and rigorously? 

Reviewer #1: Yes

Reviewer #2: N/A

Reviewer #3: (No Response)

Reviewer #4: Yes

4. Have the authors made all data underlying the findings in their manuscript fully available?

Reviewer #1: Yes

Reviewer #2: (No Response)

Reviewer #3: (No Response)

Reviewer #4: No

5. Is the manuscript presented in an intelligible fashion and written in standard English?

Reviewer #1: Yes

Reviewer #2: (No Response)

Reviewer #3: (No Response)

Reviewer #4: Yes

6. Review Comments to the Author

Reviewer #1: (No Response)

Reviewer #2: (No Response)

Reviewer #3: (No Response)

Reviewer #4: 1. The response provided by the authors to the concern about the title and "integrated analysis" approach is not accurate. The title, "Novel insights from integrated analysis," implies that multiple datasets were integrated during model development or training, which is not the case here. The study only uses GSE63060 for model training, and the additional datasets are used only for validation and subsequent analysis.

To maintain scientific accuracy, I strongly suggest revising the title to reflect the actual approach used in this study. The term "integrated analysis" should refer to the integration of multiple datasets during the model construction phase, not just the validation or post hoc analyses.

2. While the authors have addressed the similarities with previous studies, the significant overlap between this manuscript and the previous studies warrants a more detailed comparison. It would be beneficial for the authors to directly compare their results with the findings from these studies:

1. https://doi.org/10.3389/fnagi.2022.932676

2. https://doi.org/10.3389/fnagi.2023.1204530

By doing so, the authors can more clearly identify any common findings or discrepancies, strengthening their claim of novelty and providing further evidence of how their approach differs or adds to existing knowledge in the field. This would also help clarify whether their conclusions, such as those regarding cuproptosis in AD progression, are consistent with or diverge from previous studies.

7. PLOS authors have the option to publish the peer review history of their article (what does this mean? ). If published, this will include your full peer review and any attached files.

**Do you want your identity to be public for this peer review?** For information about this choice, including consent withdrawal, please see our Privacy Policy .

Reviewer #1: **Yes: ** Jai Chand Patel

Reviewer #2: No

Reviewer #3: No

Reviewer #4: No

---

## [Author Response · Author response to Decision Letter 2]

11 Apr 2025

Dear Editor and Reviewer,

We appreciate your permission to revise the manuscript again. Thank you very much for your kind, positive, and constructive comments on the manuscript. All journal requests were taken seriously. After carefully reading your comments and suggestions, we have revised and responded according to them and displayed them below:

Reviewer #4:

Q1. The response provided by the authors to the concern about the title and "integrated analysis" approach is not accurate. The title, "Novel insights from integrated analysis," implies that multiple datasets were integrated during model development or training, which is not the case here. The study only uses GSE63060 for model training, and the additional datasets are used only for validation and subsequent analysis.

To maintain scientific accuracy, I strongly suggest revising the title to reflect the actual approach used in this study. The term "integrated analysis" should refer to the integration of multiple datasets during the model construction phase, not just the validation or post hoc analyses.

Response: We sincerely appreciate your patience in explaining all this to us. We have now revised “integrated” to “comprehensive” to make our title more accurate. Please see Lines 4. We also acknowledged the potential limitations of single-dataset training in the revised discussion. Please see Lines 691-693: although our study provided a strong relationship between CAGs and AD in the peripheral immune system using multi-omics studies, the bulk and single-cell RNA data were obtained from a limited number of samples.

Q2. While the authors have addressed the similarities with previous studies, the significant overlap between this manuscript and the previous studies warrants a more detailed comparison. It would be beneficial for the authors to directly compare their results with the findings from these studies:

1. https://doi.org/10.3389/fnagi.2022.932676

2. https://doi.org/10.3389/fnagi.2023.1204530

By doing so, the authors can more clearly identify any common findings or discrepancies, strengthening their claim of novelty and providing further evidence of how their approach differs or adds to existing knowledge in the field. This would also help clarify whether their conclusions, such as those regarding cuproptosis in AD progression, are consistent with or diverge from previous studies.

Response: We sincerely thank you for highlighting the importance of contextualizing our findings within existing literature. We would like to address these points as follows:

1.Direct Comparison with Lai et al. (2022, DOI: 10.3389/fnagi.2022.932676)

Both studies confirmed the dysregulation of copper homeostasis in patients with Alzheimer's disease (AD). Unlike their cortex tissues data, we also observed elevated copper levels in peripheral blood.

While Lai et al. primarily linked copper imbalance to amyloid-β aggregation, our work explicitly establishes cuproptosis as a driver of neuronal loss via key DECAGs- -mediated dysfunction. Moreover, the content in these pictures is completely different from other studies, including the enrichment analysis, GSVA analysis in the two Clusters, machine-learning models, single-cell analysis, cellular communication, and Pet-CT results.

2. Direct Comparison with Nie et al. (2023, DOI: 10.3389/fnagi.2023.1204530)

Both studies identify cuproptosis genes as key regulators in AD progression. However, our study differs in several important respects: a. Unlike their single-cohort analysis, we employed cross-dataset validation (GSE63060 training → GSE33000/GSE122063/GSE181279 external testing) b. Unlike previous studies, we have incorporated single-cell analysis to a stronger, more targeted hypothesis regarding cuproptosis involved in AD progression. c. The number of CAGs included in this study was greater than other studies, and we employed four machine-learning methods to develop a cuproptosis-related model, which was distinguished from other work. d. Our interpretation of the results diverged from that in the referenced studies. We provided a detailed discussion on how our findings contribute to a deeper understanding of MIF pathway and intracellular communication and the potential implications for future research.

We have revised the Discussion section to ensure clarity regarding our novel findings, which also underscore our study's unique contributions while contextualizing them within existing literature. Please see Lines 673-690. We appreciate the opportunity to clarify our work’s unique position in the field and welcome further suggestions.

---

## [Decision Letter · Decision Letter 2]

7 May 2025

PONE-D-25-01107R2Novel insights from comprehensive analysis: the role of cuproptosis and peripheral immune infiltration in Alzheimer's diseasePLOS ONE

Dear Dr. Wang,

Thank you for submitting your manuscript to PLOS ONE. After careful consideration, we feel that it has merit but does not fully meet PLOS ONE’s publication criteria as it currently stands. Therefore, we invite you to submit a revised version of the manuscript that addresses the points raised during the review process.

We look forward to receiving your revised manuscript.

Kind regards,

Li Shen

Academic Editor

PLOS ONE

Journal Requirements:

Reviewers' comments:

Reviewer's Responses to Questions

**Comments to the Author**

1. If the authors have adequately addressed your comments raised in a previous round of review and you feel that this manuscript is now acceptable for publication, you may indicate that here to bypass the “Comments to the Author” section, enter your conflict of interest statement in the “Confidential to Editor” section, and submit your "Accept" recommendation.

Reviewer #4: All comments have been addressed

2. Is the manuscript technically sound, and do the data support the conclusions?

Reviewer #4: Yes

3. Has the statistical analysis been performed appropriately and rigorously? 

Reviewer #4: Yes

4. Have the authors made all data underlying the findings in their manuscript fully available?

Reviewer #4: Yes

5. Is the manuscript presented in an intelligible fashion and written in standard English?

Reviewer #4: Yes

6. Review Comments to the Author

Reviewer #4: Thank you to the authors for addressing the comments. However, I would kindly suggest improving the quality of Figures 8, 9, and 11, as they currently appear unclear. Enhancing their quality would significantly improve clarity and readability.

7. PLOS authors have the option to publish the peer review history of their article (what does this mean? ). If published, this will include your full peer review and any attached files.

**Do you want your identity to be public for this peer review?** For information about this choice, including consent withdrawal, please see our Privacy Policy .

Reviewer #4: No

---

## [Author Response · Author response to Decision Letter 3]

11 May 2025

Dear Editor and Reviewer,

We sincerely appreciate your permission to revise the manuscript once again. Thank you very much for your kind, positive, and constructive feedback, which has been invaluable in improving our work. We have carefully reviewed all the journal’s requests and have given them our utmost attention. After thoroughly considering your comments and suggestions, we have meticulously revised the manuscript accordingly. Below, we have provided a detailed response to each point, outlining the changes made to address your recommendations.

Response to Academic Editor:

Response: Thank you for your kind and constructive reminder of our work on referencing. Based on your comments, we have carefully reviewed the reference list and confirmed that all articles in the References have met the criteria for ‘cannot be retracted’ status. We sincerely appreciate your valuable comments.

Response to Reviewer #4:

Q1. Thank you to the authors for addressing the comments. However, I would kindly suggest improving the quality of Figures 8, 9, and 11, as they currently appear unclear. Enhancing their quality would significantly improve clarity and readability.

Response: Thank you for your insightful comments on the quality of Figures 8, 9, and 11, which will improve the integrity of our study greatly. We have Enhanced the quality of these figures and displayed them in PDF format to improve clarity and readability of our study based on your advice. We hope these revisions adequately address your concerns. Thank you once again for your thoughtful review.

---

## [Decision Letter · Decision Letter 3]

21 May 2025

Novel insights from comprehensive analysis: the role of cuproptosis and peripheral immune infiltration in Alzheimer's disease

PONE-D-25-01107R3

Dear Dr. Wang,

We’re pleased to inform you that your manuscript has been judged scientifically suitable for publication and will be formally accepted for publication once it meets all outstanding technical requirements.

Kind regards,

Li Shen

Academic Editor

PLOS ONE

Additional Editor Comments (optional):

Reviewers' comments:

Reviewer's Responses to Questions

**Comments to the Author**

1. If the authors have adequately addressed your comments raised in a previous round of review and you feel that this manuscript is now acceptable for publication, you may indicate that here to bypass the “Comments to the Author” section, enter your conflict of interest statement in the “Confidential to Editor” section, and submit your "Accept" recommendation.

Reviewer #3: All comments have been addressed

2. Is the manuscript technically sound, and do the data support the conclusions?

Reviewer #3: (No Response)

3. Has the statistical analysis been performed appropriately and rigorously? 

Reviewer #3: (No Response)

4. Have the authors made all data underlying the findings in their manuscript fully available?

Reviewer #3: (No Response)

5. Is the manuscript presented in an intelligible fashion and written in standard English?

Reviewer #3: (No Response)

6. Review Comments to the Author

Reviewer #3: (No Response)

7. PLOS authors have the option to publish the peer review history of their article (what does this mean? ). If published, this will include your full peer review and any attached files.

**Do you want your identity to be public for this peer review?** For information about this choice, including consent withdrawal, please see our Privacy Policy .

Reviewer #3: No

---

## [Editor Report · Acceptance letter]

PONE-D-25-01107R3

PLOS ONE

Dear Dr. Wang,

I'm pleased to inform you that your manuscript has been deemed suitable for publication in PLOS ONE. Congratulations! Your manuscript is now being handed over to our production team.

Kind regards,

on behalf of

Dr. Li Shen

Academic Editor

PLOS ONE